# HYBRID MODEL COLLABORATION FOR SIGN LANGUAGE TRANSLATION WITH VQ-VAE AND RAG-ENHANCED LLMS

## ABSTRACT

Data shortages and the phonetic disparity between sign and spoken languages have historically limited the quality of sign language translation. On another front, endowed with substantial prior knowledge, large language models perform exceptionally well across diverse tasks, significantly diminishing the demand for domain-specific training data. Building on these foundation, this paper presents VRG-SLT, an innovative framework that translates sign language into spoken language, facilitating communication between signing and non-signing communities. In practice, VRG-SLT utilizes a hierarchical VQ-VAE to convert continuous sign sequences into discrete representations, referred as sign codes, which are subsequently aligned with text by a fine-tuned pre-trained language model. Additionally, retrieval-augmented generation (RAG) is employed to extend and enhance the language model, producing more semantically coherent and precise spoken text. Featuring a hierarchical VQ-VAE and pre-trained large language models, VRG-SLT demonstrates state-of-the-art performance. It excels on modish benchmarks like How2Sign and PHOENIX-2014T. Moreover, the incorporation of additional factual knowledge through RAG further improves the accuracy of the generated text. The implementation code will be released.

## 1 INTRODUCTION

Characterized by unique linguistic traits, sign languages play a crucial role in communication among deaf individuals (Padden & Humphries, 1988; Stokoe Jr, 2005; Glickman & Hall, 2018). Unlike spoken languages, they rely on visual cues like gestures, body movements, facial expressions, and eye movements to convey semantic information (Liddell & Johnson, 1989; Johnson & Liddell, 2011; Sandler, 2012). Sign language translation (SLT) involves converting sign gestures from video clips into spoken descriptions (Camgöz et al., 2018; 2020; Zhou et al., 2021; 2022; De Coster et al., 2023), facilitating communication freedom and accessibility of information for both sign and non-sign language users. In practice, SLT highlights its versatility and significant value across various scenarios (Harris et al., 2009), such as public service broadcasts, and personal assistants, *etc*.

Building effective and accurate sign language translation systems commonly encounters the following obstacles: 1) **Data scarcity**: The collection of sign language data is particularly challenging, owing to its limited user population and the considerable costs and complexities of data gathering and annotation. For instance, the How2Sign dataset (Duarte et al., 2021) contains only $30,000$ pairs, hampering effective model optimization. 2) **Unique syntax**: Sign language, inherently distinct from spoken language, possesses its own grammar, word formation, and lexicon. These differences, especially in word order, make transcription between the two languages complex. 3) **Multimodal contexts**: Sign language is a multimodal form of communication that combines manual and non-manual actions, such as facial expressions and body postures, to convey detailed and precise information. These traits culminate in numerous signs that are visually similar but distinct in their semantic implications.

Previous research generally divides SLT into two distinct tasks (Chen et al., 2022): **Sign2Notation (or sign language recognition, SLR)**, the conversion of sign language videos to lexical represen-

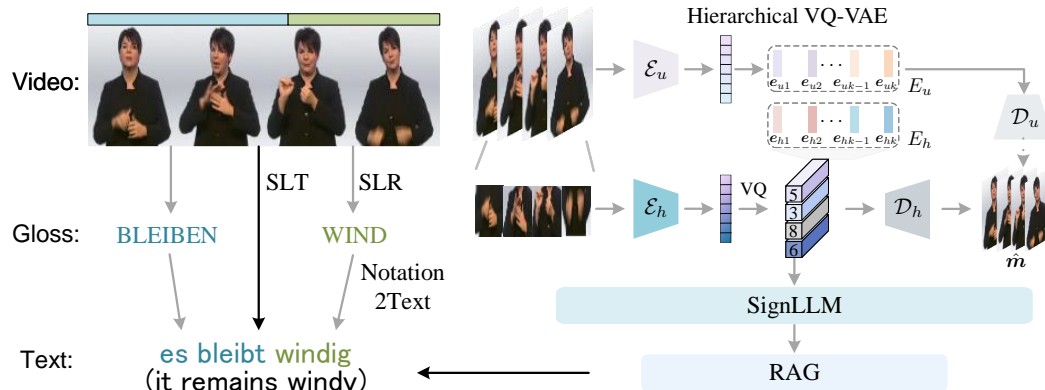

Figure 1: We propose VRG-SLT, a hybrid collaborative sign language translation framework that integrates a hierarchical VQ-VAE (sign-tokenizer) with the pretrained language model FLAN-T5. Furthermore, we employ a RAG strategy to calibrate and refine the initial outputs (§1).

tations (*e.g.*, Glosses*); and **Notation2Text**, the translation of these representations into the target text language, a process akin to machine translation but dealing with complex, multimodal sign inputs (Fig. 1). SLR (Padden, 2016; Cui et al., 2017; Pu et al., 2019; Li et al., 2020b; Zhou et al., 2022) endeavors to decipher successive signs as discrete gloss lexicon. However, it disregards the differing grammar and linguistic structure of sign language from spoken language. As a result, translations often lack semantic coherence and sentence fluidity. Notation2Text (Camgöz et al., 2018; 2020; Li et al., 2020a; Duarte et al., 2021; Zhou et al., 2021; 2022) struggles to fully capture non-verbal elements of sign language like facial expressions. It relies on extensive bilingual corpora, posing a significant challenge for resource-scarce sign languages. SLT seeks to convert sign language videos into spoken sentences, ensuring accuracy and comprehensibility by accounting for grammatical and word order differences. With the rise of deep learning, SLT has shifted from traditional feature engineering to adopting various neural network approaches, such as convolutional networks, LSTM (Hochreiter & Schmidhuber, 1997) and transformers (Vaswani et al., 2017), to enhance translation accuracy. Recent efforts treat sign language translation as a unified task and introduce domain-specific knowledge or extensive auxiliary training data (Chen et al., 2022; Zhao et al., 2023; Rust et al., 2024), yet the accuracy and generalization remain inadequate for real-world applications. Recently, large language models (LLMs) (Devlin et al., 2019; Yang et al., 2019; Brown et al., 2020; Lan et al., 2020; Lewis et al., 2020a) exhibit strong comprehension and possess extensive prior knowledge, lessening their dependency for large-scale data. LLMs are highly effective across various contexts, maintaining accuracy and robustness, showcasing their versatility in various applications. This can greatly benefit the SLT field, known for limited corpora availability. However, despite early trials (Wong et al., 2024), the application of LLMs in sign language translation is still not sufficiently explored.

In this paper, we concentrate on incorporating sign language gestures into large language models to translate them into spoken text (Fig. 1). To achieve this, we propose VRG-SLT, a two-stage pipeline. Initially, a sign-specific VQ-VAE (sign-tokenizer) quantizes raw sign segments into discrete codes. These codes are then converted into spoken sentences by FLAN-T5. Presently, the prevailing LLMs are text-centric and lack the capability to directly translate sequences of sign language into text. How to jointly train sign videos with text? The answer lies in developing a sign-tokenizer that embeds sign clips into discrete tokens and aligns them with text during the finetuning of LLMs. Our approach integrates insights from VQ-VAE-2 (Razavi et al., 2019) and text-to-motion technologies (Zhang et al., 2023b; Jiang et al., 2023), utilizing a hierarchical VQ-VAE (van den Oord et al., 2017) and a pretrained language model FLAN-T5 (Chung et al., 2024) to efficiently convert signs into spoken sequences. Furthermore, the translations are enhanced by a retrieval-augmented generation (RAG) (Lewis et al., 2020b) strategy, further improving the performance of VRG-SLT beyond preliminary results. In particular, sign-tokenizer compresses sign clips by encoding them into latent representations, which are then quantized into discrete codes and stored as indices in a codebook, referred to as "sign vocabulary" (van den Oord et al., 2017). Traditional VQ-VAEs prioritize upper body motion, making them less suitable for sign language that emphasizes hand and torso features. Inspired by VQ-VAE-2,

---

*Glosses are the practice of describing sign language actions with written words to express their meanings. For instance, the sign for a dog may be denoted with the gloss 'DOG'.

we design sign-tokenizer as a two-level network: the top level captures body information, such as the motion trajectories of the shoulders and elbows, while the bottom level focuses on modeling the movements of hands. The body features is then infused into the bottom level's hand information for precise sign reconstruction. This hierarchical and multi-scale representation allows the sign-tokenizer to detect and capture details across different levels of granularity. A "text-sign dictionary" can be constructed from the sign and conventional text corpus. Subsequently, FLAN-T5 is finetuned to jointly learn and bridge the syntax and grammar of the "text-sign language". One notable limitation is that LLMs such as T5 may lack sufficient or in-depth domain knowledge, tending to produce inaccurate or unrealistic responses (known as 'hallucination'). Thus, to rectify incorrect answers, we adopt RAG strategy that retrieves pertinent knowledge and polishes the initial translations. The hybrid collaboration integrates sign-tokenizer for encoding sign motions, FLAN-T5 for text generation, and RAG to enhance accuracy and cultural relevance. Each component contributes its strengths, working synergistically to tackle complex sign language translation challenges.

We are pioneering the integration of hierarchical VQ-VAE and LLMs into SLT, bolstered by RAG for enhanced translation. The main contributions are summarized as follows:(1) A collaborative hybrid model, VRG-SLT, is introduced for sign language translation, where sign movements are treated as a unique language and combined with LLMs for joint training with text. (2) A sign-tokenizer, which captures both overall upper body and hand trajectory characteristics, is presented. By utilizing a hierarchical structure, it can adeptly handle intricate detailed complexities and diverse contextual movements. (3) RAG strategy is integrated into VRG-SLT, enabling the retrieval and combination of relevant knowledge for more accurate and content-rich output. VRG-SLT notably surpasses competitors on benchmarks such as How2Sign (Duarte et al., 2021) and PHOENIX-2014T (Camgöz et al., 2018), including those employing semi-supervised learning. For instance, VRG-SLT achieves remarkable gains on the How2Sign dataset, with ROUGE and BLEU-1 scores increasing by 2.23 and 4.34, respectively. Our code will be released.

## 2 RELATED WORK

**Sign Language Understanding and Translation** aims to precisely recognize and explain sign language components such as the shapes, positions, and movements, translating them into equivalent verbal language. Isolated SLR and the more challenging continuous SLR are two fundamental tasks for understanding sign language. One aims to identify single annotated word labels in short video clips (Albanie et al., 2020; Li et al., 2020b), while the other seeks to convert continuous sign videos into gloss sequences using only weak sentence-level annotations (Cui et al., 2017; Koller et al., 2020; Pu et al., 2019; Zhou et al., 2022). While some previous studies (Padden, 2016) equate SLR with SLT, the former merely classifies signs, neglecting their grammatical and morphological structures into spoken language. Jiang et al. (2021) propose a fresh multimodal framework featuring a globally integrated model for skeleton-aware multimodal learning in discrete SLR. To date, SLR has been simplified to a basic gesture recognition issue, thereby overlooking the linguistic aspects of sign language and presuming a direct correlation between sign and spoken words. While the encoder-decoder network in NMT boosts translation, it grapples with an info bottleneck from condensing source sequences to fixed vectors and managing long-term dependencies across source and target texts. Generally, SLR serves as an intermediate step in the translation process, annotating sign language videos before converting them into spoken language through a sequence-to-sequence method (Notation2Text) (Camgöz et al., 2018; 2020; Li et al., 2020a; Duarte et al., 2021; Zhou et al., 2021; 2022). For instance, Camgöz et al. (2020) integrate the training of SLT to regularize the translation encoder. Zhou et al. (2021) introduce a data augmentation approach that uses annotations as pivots to back-translate text into visual features. Cico (Cheng et al., 2023) models the relationship between signs and text from a cross-lingual retrieval perspective. Chen et al. (2022) develop a unified framework for SLT, dividing it into visual and linguistic modules bridged through a visual-linguistic mapper for training. Influenced by action recognition (Ji et al., 2013; Tran et al., 2015; Arnab et al., 2021), some studies (Camgöz et al., 2017; Niu & Mak, 2020; Cheng et al., 2020; Min et al., 2021; Hao et al., 2021) explore directly modeling RGB videos to understand sign language. However, these methods still struggle with sentence-level translation. Our approach utilizes sign-tokenizer to treat raw sign motions as equivalent to textual words for whole-sentence translation, co-training with spoken text to transcend the linguistic barriers.

**VQ-VAE**, an unsupervised learning technique, excels in compressing and reconstructing high-fidelity images, videos, and audio by mapping them into a lower-dimensional latent space (van den Oord

et al., 2017). Its recent applications extend to realistic image and video generation with GANs, diverse vocal representations in speech processing, and feature extraction in unsupervised learning (Esser et al., 2021; Chang et al., 2022; Lee et al., 2022; Zheng et al., 2022). The classic VQ-VAE structure consists of an encoder, a vector quantizer, and a decoder. VQ-VAE encodes the input into a discrete latent representation. This is achieved by mapping the encoder outputs to a nearest vector in a predefined, learnable codebook. The process involves several key steps: **(1) Encoding**: The encoder converts the raw input data into a latent representation. **(2) Quantization**: The core of VQ-VAE lies in its vector quantization, where the continuous representation of the encoder is mapped to the nearest code in the codebook, improving model performance on complex data distributions and sample fidelity. **(3) Reconstruction:** The quantized vectors are then passed to the decoder, which attempts to reconstruct the original input data. The training objective of VQ-VAE includes a reconstruction loss to minimize the difference between the input and the reconstructed output. In text-to-motion generation (Zhang et al., 2023b; Jiang et al., 2023), VQ-VAE delivers compelling outcomes in semantic coherence and motion precision. Despite VQ-VAE achieving accuracy comparable to that of continuous vector counterparts, it also exhibits the typical autoencoder drawback of image blurring. The following VQ-VAE-2 (Razavi et al., 2019) employs a multi-level hierarchical structure to produce images of superior quality while maintaining diversity and preventing mode collapse. This paper, inspired by VQ-VAE-2, carefully crafts a sign-tokenizer that uses top and bottom level quantizers to model the upper body and hand regions, thereby capturing more detailed and comprehensive motion trajectories. To our knowledge, this is the pioneering effort to utilize multi-level VQ-VAE specifically for the realm of sign language translation.

**Large Language Models** revolutionize natural language processing with their ability to generate human-like text (Peters et al., 2018; Devlin et al., 2019; Dong et al., 2019; Liu et al., 2019; Lan et al., 2020). Predominantly utilized for text generation, language translation, and automated customer service, LLMs stand out due to their extensive pre-training on diverse data. Recent improvements have scaled the model up, boosting its coherence and contextual relevance, particularly in chatbots and creative writing (Clark et al., 2020; Sun et al., 2020; Lewis et al., 2020a). Additionally, GPT is branching into multimodal applications, like AI art and data analysis. Current research focuses on improving understanding, reducing biases, and enhancing computational efficiency, cementing GPT's role as a pivotal AI tool. T5 model (Raffel et al., 2020) employs a unified text-to-text framework, transforming diverse NLP tasks into text generation issues. Moreover, T5, trained on diverse language corpora, exhibits extensive prior knowledge and robust cross-lingual capabilities. FLAN-T5 (Chung et al., 2024) boosts the multi-task proficiency of T5 by natural language training and command response refinement. Our framework uniquely merges FLAN-T5 and VQ-VAE-2, equating sign tokens with text tokens as "word", thus boosting cross-lingual alignment in full-sentence translation.

**Retrieval-Augmented Generation** is a language enhancement technique that merges information retrieval with generation models. RAG begins by pulling relevant information from a knowledge base and fuses it with a generative model to produce more precise and comprehensive text output (Lewis et al., 2020b; Mallen et al., 2023; Shi et al., 2023; Morris et al., 2023; Asai et al., 2024). It usually provides several benefits: (1) RAG substantially improves the fidelity of generated responses, especially in scenarios requiring accurate answers to factual queries. (2) It diminishes the frequency of hallucinations by the generation model. By integrating RAG, we utilize a knowledge base to enhance the translation of sign language into text.

## 3 METHOD

We propose VRG-SLT, a framework for translating sign language into spoken text. As illustrated in Fig. 2, VRG-SLT comprises a sign-tokenizer, a sign-aware language model, and a RAG module. Sign-tokenizer (§3.1) employs a hierarchical VQ-VAE-2 to encode raw sign sequences into discrete codes in a codebook. These codes, along with spoken texts, establish a new "text-sign dictionary" for cross-lingual learning. Next, the sign-aware large language model SignLLM (§3.2) focuses on aligning sign motions with corresponding textual descriptions. Furthermore, RAG (§3.3) accesses relevant knowledge to refine output text and alleviate hallucinations. In practice, sign-tokenizer consists of 2 sign encoders, $\mathcal{E}_u$ and $\mathcal{E}_h$, and a sign decoder $\mathcal{D}$. Sign-tokenizer first maps a sign motion sequence $\boldsymbol{m}^{1:M}$ of $M$ frames into $L$ motion codes $\boldsymbol{e}^{1:L}$, and decodes $\boldsymbol{e}^{1:L}$ back into a reconstructed motion sequence $\hat{\boldsymbol{m}}^{1:M} = \mathcal{D}(\boldsymbol{e}^{1:L})$. Here, $L = M/l$, $l$ denotes the temporal downsampling rate. The goal of SignLLM is to generate corresponding verbal text $\hat{\boldsymbol{t}}^{1:N}$ with $N$ words conditioned on the sign code sequence $\boldsymbol{e}$, denoted as $\hat{\boldsymbol{t}}^{1:N} = SignLLM(\boldsymbol{e}^{1:L})$.

Figure 2: VRG-SLT mainly comprises a sign-tokenizer (§3.1) and a sign-aware language model, Sign-LLM (§3.2). The sign-tokenizer encodes sign actions into a *sign codebook* and, together with the text tokenizer, creates a unified vocabulary $V$. Using SignLLM, we perform joint learning of sign and spoken languages for sign language translation. The two encoders of the sign-tokenizer encode global body movements and detailed hand features, respectively, achieving a comprehensive and precise understanding of sign motion. Finally, we refine the initial output using a RAG strategy (§3.3).

## 3.1 SIGN TOKENIZER

To begin with, we revisit the workflow of VQ-VAE. VQ-VAE typically consists of three main components: an encoder, a quantizer, and a decoder. The process begins with the encoder converting input data (*e.g.*, images or audio) into a latent representation. Following this, the quantizer maps this representation to a set of nearest discrete codes. These codes are then forwarded to the decoder for input reconstruction. In this paper, sign-tokenizer, designed to represent sign language in discrete codes, is pre-trained based on the principles of VQ-VAE (van den Oord et al., 2017; Siyao et al., 2022; Zhang et al., 2023b). The quantizer assigns $z$ to the nearest vector $e_i$ in the codebook. Thus, sign motions $m$ can be represented as an integer index $k$: $0 \leq k < K$, with a vocabulary size $K$. The sign-tokenizer, featuring a hierarchical architecture with two encoders $\mathcal{E}_u$, $\mathcal{E}_h$ and a decoder $\mathcal{D}$, is tailored to capture sign characteristics comprehensively. The encoders and quantizers generate highly informative discrete sign codes, while the decoder reconstructs these codes into sign sequences $\hat{m}^{1:M}$. Sign-tokenizer can effectively represent sign movements as code sequences, facilitating the integration of sign and spoken sentences in SignLLM. Then, the sign-tokenizer applies quantizers to the upper ($z_u$) and lower ($z_h$) vectors for each input. The quantized representations, $e_u$ and $e_h$, are utilized by the VQ-VAE to establish a joint probability density for overarching semantic features $p_u$ and the conditional probability density for detailed local mappings $p_h$. The generation process concludes by sampling quantized codebook vectors from $p_u$ for global consistency and $p_h$ for local detail, which are then fed into the decoder $\mathcal{D}$ to generate reconstructed sign sequences.

Specifically, both the sign encoders first applies 1D convolutions to the frame-wise sign motions $m^{1:M}$ along the time dimension, generating latent vectors $z_u$ and $z_h$. These latent vectors are then discretized into codebook entries $e_{uk}$ and $e_{hk}$. Both the codebook $E_u = \{e_{uk}|k = 1, \ldots, K\}$ and $E_h = \{e_{hk}|k = 1, \ldots, K\}$ contain $K$ embedding vectors, each of dimension $d$. The quantizers $Q_u$ and $Q_h$ maps each vector with its nearest codebook entry in $E_u$ and $E_h$, respectively (Eq. 1). After quantization, the sign decoder $\mathcal{D}$ projects $e_h^{1:L}$ back to raw motion space as $\hat{m}^{1:M}$.

$$
\begin{aligned}
e_{uk} = Q_u(\hat{z}_u) &:= \arg\min_{e_{uk} \in E_u} \|\hat{z}_u - e_{uk}\|_2 \,; \\
e_{hk} = Q_b(\hat{z}_h) &:= \arg\min_{e_{hk} \in E_h} \|\hat{z}_h - e_{hk}\|_2 \,.
\end{aligned}
\tag{1}
$$

**VQ-VAE Learning.** VQ-VAE employs a unique learning strategy that updates the embeddings in the codebook by using an exponential moving average of the encoder outputs, which helps in stabilizing the training process. We train our motion tokenizer using the method outlined in (Guo et al., 2022b; Zhang et al., 2023b) to synchronize the vector space of the codebook , with three distinct loss functions for optimization. The codebook loss applies only to codebook variables, drawing the selected codebook vector closer to the encoder outputs. The commitment loss applies solely to

Figure 3: Following the classic RAG workflow, we first integrates a retrieval step into the generative model, pulling relevant documents from knowledge base to inform and refine the initial output (§3.3).

the encoder weights, ensuring the encoder output remains close to the chosen codebook vector to minimize frequent shifts between code vectors. The overall objective is described in Eq. 2, where $e_u$ and $e_h$ represent the quantized code for training sample $m$. $sg$ denotes a stop-gradient operation that prevents gradients from flowing into its argument. $\beta_1$ and $\beta_2$ is a hyperparameter that controls resistance to changes in the encoder's code output.

$$\mathcal{L}_V = \|m - \mathcal{D}(e_h)\|_2 + \|sg[\mathcal{E}_u(m)] - e_u\|_2 + \\ \|sg[\mathcal{E}_h(m)] - e_h\|_2 + \beta_1\|sg[e_u] - \mathcal{E}_u(m)\|_2 + \beta_2\|sg[e_h] - \mathcal{E}_h(m)\|_2. \quad (2)$$

We empirically set $\beta_1$ and $\beta_2$ to 1, respectively. To enhance the quality of the generated motion, we also employ velocity regularization, and codebook reset (Razavi et al., 2019). Further details on the architecture and training of the sign tokenizer are provided in the supplement.

## 3.2 SIGNLLM

With the sign-tokenizer, sign motions $m^{1:M}$ can be converted into a sign token sequence $e^{1:L}$, which facilitates joint representation with similar text embeddings in language models (Kudo & Richardson, 2018; Raffel et al., 2020; Ouyang et al., 2022). The unified vocabulary allows for simultaneous learning from sign and spoken languages, supporting hybrid collaboration between the sign tokenizer and SignLLM. Unlike previous text-to-motion approaches (Guo et al., 2022b; Chen et al., 2023; Zhang et al., 2023b) that adopt separate modules for text and sign sequence processing, our approach aims to integrate text and sign motion processing in a unified manner. To achieve this, we merge the original text vocabulary $V_t = \{v_t\}$ with the sign vocabulary $V_m = \{v_m\}$, which preserves the order in our sign codebook $E_h$. The sign vocabulary $V_m$ contains special tokens like boundary indicators, such as </sos> and </eos> for the start and end of sign, respectively. With the unified text-sign vocabulary $V = \langle V_t, V_m \rangle$, we can handle sign and text data in a general format, where both input and output "words" are tokens drawn from the same vocabulary. These tokens can represent spoken language, sign motion, or a combination of both. As a result, our method enables flexible representation of diverse sign-related outputs within a single $SignLLM$.

We combine the sign codes and prompts into the input sequence $x_s$, each element of which belongs to the unified vocabulary $V$. Then, $x_s$ serves as the context or conditioning for SignLLM to produce the output spoken text $\hat{t}$. As depicted in Fig. 2, the source tokens $x_s$ enter SignLLM encoder, and SignLLM decoder predicts the probability distribution of the next token at each step, $p_\theta(t \mid x_s) = \prod_i p_\theta\left(t^i \mid t^{<i}, x_s\right)$. The objective is to maximize the log-likelihood as follows:

$$\mathcal{L}_{LM} = -\sum_{i=0}^{L-1} \log p_\theta\left(t^i \mid t^{<i}, x_s\right). \quad (3)$$

By optimizing this objective, VRG-SLT can capture the underlying patterns and relationships from data distribution, thereby facilitating the accurate predict of target tokens. During inference, the target tokens are sampled recursively from the predicted distribution $p_\theta\left(\hat{t}^i \mid \hat{t}^{<i}, x_s\right)$ until the end token (*i.e.*, ) is reached. Since each token in the target sequence is generated based on both preceding tokens and the original input, SignLLM effectively maintains semantic consistency. FLAN-T5 is proficient in multi-task fine-tuning, offering strong adaptability across diverse natural language processing tasks. Thus, we chose FLAN-T5 as the backbone for large model.

## 3.3 RAG

RAG (Lewis et al., 2020b) has become a paradigm in the LLM field for enhancing the capabilities of generative tasks. Unlike purely generative models, RAG decreases errors and irrelevant outputs by incorporating relevant background information. Specifically, RAG incorporates a distinct initial step.

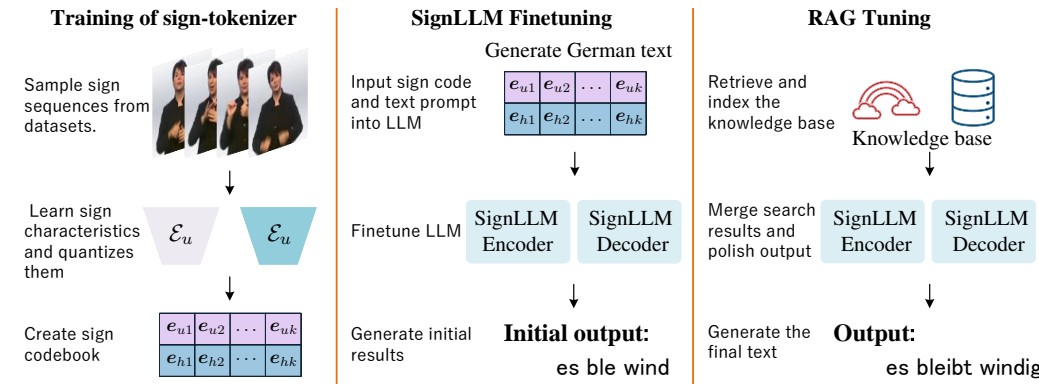

Figure 4: Training Scheme. VRG-SLT comprises three steps (§3.4): First, sign-tokenizer learns a codebook for discrete sign representations. Next, we train the language model SignLLM using a mix of spoken and sign data to understand the semantic coupling between text and sign motion. Finally, we polish the initial output using RAG.

The LLM first queries external data sources for relevant information. After gathering the necessary knowledge, it then proceeds to generate text or answer questions. This strategy not only guides the generation phase but also uses retrieved evidence to enhance the accuracy and relevance of responses, reducing content errors known as hallucinations. The workflow consists of retrieving documents relevant to an input from a large corpus, followed by generating the final response or content based on these documents and the original query. Following the classic RAG workflow, we also engage in the steps of indexing, retrieving, and generation:

- Indexing: Initially, documents are converted into vectors and stored within an indexed database. Then,query data is segmented into manageable chunks and transformed into vectors using a well-balanced embedding model. This process enhances similarity comparisons and supports efficient search by storing these vectors and their associated text in an index.

- Retrieving: When a query is received, the system transcodes the query into vectors using the initial encoding model, as shown in Fig. 3. It then calculates similarity scores with the indexed vectorized chunks, retrieving the top-3 chunks with the highest similarity for extended context analysis.

- Generation: The model synthesizes the query and retrieved knowl edge into a prompt to generate responses. Responses may vary by different motion codes or prompt text.

## 3.4 TRAINING PROCEDURES

FLAN-T5, originally pre-trained with a text-based vocabulary $V_t$, is aligned with sign language through the sign-specific vocabulary $V_m$. Our training steps consist of three stages (Fig. 4): (1) training the sign-tokenizer to represent signs with discrete codes; (2) finetuning on sign language to bridge sign motion and language; and (3) tuning the output with RAG. We provide the pseudocode for each stage in the appendix (§B).

**Training of Sign-tokenizer.** Initially, sign-tokenizer is trained using the objective defined in Eq. 2, enabling any sign sequence $m^{1:M}$ to be represented as a sequence of motion tokens, which integrates seamlessly with textual information. After optimization, the sign-tokenizer remains unchanged throughout the rest of the pipeline.

**SignLLM Finetuning.** The SignLLM are then trained and fine-tuned on the unified text-sign vocabulary $V = <V_t, V_m>$. We utilize existing sign language datasets (such as How2Sign (Duarte et al., 2021) and PHOENIX-2014T (Camgöz et al., 2018)) as a foundation to create a guided sign-action dataset. As explored in prior works (Devlin et al., 2019; Radford et al., 2019; Raffel et al., 2020; Ouyang et al., 2022), we also adopt an objective inspired by (Raffel et al., 2020) where 15% of input tokens are randomly replaced with a sentinel token. The target sequence is then constructed by extracting the dropped-out spans, delimited by the same sentinel tokens, with an additional sentinel token marking the end of the sequence. We establish the relationship between motion and language using paired text-sign datasets (Guo et al., 2022a; Plappert et al., 2016). Through training, our model

is intended to comprehend the relationship between text and motion. For example, the prompt might say: "Generate English text: <sign_tokens>" or "Generate German text: <sign_tokens>". Here, <sign_tokens> refers to the token form of sign codes from sign-tokenizer.

**RAG Tuning.** We utilize the SQuAD database (Devlin et al., 2019) for general knowledge expansion and ECMWF (Hersbach et al., 2020) for weather data. After retrieving relevant information, we combine the initial output with the retrieved knowledge and input them into an open-source large model for refinement. BERT (Devlin et al., 2019) is employed for retrieving.

# 4 EXPERIMENTS

**Dataset.** We assess performance on **How2Sign** (Duarte et al., 2021) and **PHOENIX-2014T** (Camgöz et al., 2018) datasets, which are prevailing benchmarks in sign language understanding.

- How2Sign is a comprehensive multimodal American Sign Language dataset, comprising approximately 80 hours of sign language videos with corresponding annotations. It consists of $31,164$, $1,740$, and $2,356$ sign-video-text triplets for training, validation, and testing, respectively.

- PHOENIX-2014T, a German Sign Language (DGS) dataset, consists of weather forecast segments extracted from TV broadcasts. Each video is accompanied by detailed sign language annotations and corresponding German spoken text. The dataset is split into $7,096$ train, $519$ validation, and $642$ test examples, respectively.

**Evaluation Metric.** Drawing from previous research (Camgöz et al., 2018; 2020; Zhou et al., 2021; 2022), we utilize the widely used ROUGE (Lin, 2004) and BLEU (Papineni et al., 2002) metrics to evaluate precision and fluency of translated content. The assessments are conducted by comparing machine-produced texts to human reference texts. Focused on accuracy, BLEU measures translation quality through the overlap of n-grams between the machine output and reference texts. However, it might not adequately capture the fluency and semantic precision of the translation. ROUGE evaluates content coverage through the overlap between translation texts and human reference materials. In summary, ROUGE primarily assesses the overlap between generated and reference text, focusing on recall, whereas BLEU emphasizes precision through n-gram matching.

**Implementation Details.** Sign language motion data is represented in the form of keypoints. The codebook of sign-tokenizer consists of 512 vectors, each of dimension 1024. The encoder applies a temporal downsampling rate of 4, merging every four frames into a single sign code to effectively capture fundamental dynamic features (Jiang et al., 2023). We utilize FLAN-T5-base (Raffel et al., 2020) as the underlying architecture for our language model. Moreover, all our models employ the AdamW (Loshchilov & Hutter, 2019) optimizer for training. The sign-tokenizer are trained utilizing a $10^{-3}$ learning rate and a 512 mini-batch size, while our SignLLM have a $2 \times 10^{-4}$ learning rate for the finetuning stage and a 32 mini-batch size. Sign-tokenizer and SignLLM undergo 100k and 200k training iterations, respectively. BERT is applied for querying and retrieving relevant knowledge, producing higher-quality outputs based on the initial output from SignLLM and the content of retrieved documents. All models are trained on 8 Nvidia GeForce RTX 4090 GPUs. We will release our code to ensure reproducibility.

## 4.1 COMPARISONS WITH SOTA METHODS

VRG-SLT treats sign motion as a unique language, incorporating a hierarchical VQ-VAE and SignLLM, with further accuracy enhancements through RAG. We utilize the $220M$ pre-trained *Flan-T5-Base* (Raffel et al., 2020; Chung et al., 2024) model as the backbone, finetuning it through the unified codebook (§3.2) for all subsequent comparisons. The results are calculated with a $95\%$ confidence interval from 10 repeated runs. The results in Table 1 indicate that VRG-SLT delivers strong performance across all metrics, demonstrating its cross-language learning ability and semantic consistency [†]. Notably, it achieves a BLEU-4 score of 30.17 on PHOENIX-2014T dataset, exceeding the nearest competitor by 1.70 points, and scores 53.92 in ROUGE, surpassing others by 1.81 points. These outcomes highlight VRG-SLT can more effectively decode and render sign language nuances into accurate and fluid translations. Our model prioritizes contextual coherence, leveraging LLM's strong capability for context modeling to produce coherent, semantically consistent

---

[†]Some comparisons are showcased on our webpage: https://vrg-slt.github.io/VRG-SLT-demos

Table 1: Compared with state-of-the-art methods on How2Sign and PHOENIX-2014T. Methods marked with an asterisk (*) first perform SLR and then Notation2Text (§4.1).

| Methods | How2Sign | | | | | PHOENIX-2014T | | | | |
|---|---|---|---|---|---|---|---|---|---|---|
| | ROUGE↑ | BLEU-1↑ | BLEU-2↑ | BLEU-3↑ | BLEU-4↑ | ROUGE↑ | BLEU-1↑ | BLEU-2↑ | BLEU-3↑ | BLEU-4↑ |
| SL-Luong (Camgöz et al., 2018) | 18.75 | 19.46 | 9.53 | 4.67 | 3.21 | 31.80 | 32.24 | 19.03 | 12.83 | 9.58 |
| TSPNet-Joint (Li et al., 2020a) | 16.84 | 17.93 | 11.71 | 6.59 | 4.07 | 34.96 | 36.10 | 23.12 | 16.88 | 13.41 |
| SL-Transf (Camgöz et al., 2020) | 21.92 | 24.74 | 13.66 | 8.20 | 5.18 | 37.31 | 46.61 | 33.73 | 26.19 | 21.32 |
| STMC-T (Zhou et al., 2022) | 25.40 | 29.38 | 15.27 | 8.68 | 6.05 | 46.65 | 46.98 | 36.09 | 28.70 | 23.65 |
| SIGN2GPT (Wong et al., 2024) | 25.83 | 28.82 | 14.84 | 8.41 | 5.93 | 48.90 | 49.54 | 35.96 | 28.83 | 22.52 |
| TIN-Trans* (Zhou et al., 2021) | 26.33 | 28.20 | 15.02 | 9.24 | 6.28 | 49.54 | 50.80 | 37.75 | 29.72 | 24.32 |
| SignBERT+ (Hu et al., 2023) | 28.35 | 29.06 | 15.71 | 9.60 | 6.84 | 50.63 | 52.01 | 39.19 | 31.06 | 25.70 |
| SLRT (Chen et al., 2022) | 31.27 | 30.10 | 18.13 | 10.43 | 7.98 | 52.65 | 53.97 | 41.75 | 33.84 | 28.39 |
| SLTUNET (Zhang et al., 2023a) | 31.15 | 31.27 | 18.02 | 10.36 | 8.19 | 52.11 | 52.92 | 41.76 | 33.99 | 28.47 |
| **VRG-SLT (Ours)** | $33.38_{\pm.02}$ | $35.61_{\pm.04}$ | $20.35_{\pm.03}$ | $13.12_{\pm.06}$ | $8.53_{\pm.03}$ | $53.92_{\pm.05}$ | $55.74_{\pm.01}$ | $43.31_{\pm.01}$ | $36.59_{\pm.05}$ | $30.17_{\pm.06}$ |

sentences. Leveraging the strong capabilities of LLM in context modeling, VRG-SLT prioritizes contextual coherence to generate semantically consistent sentences. This is reflected in its ROUGE scores, which measure how well the translated text covers the reference text vocabulary. Non-verbal information such as expressions and body language is crucial in sign language for conveying complete meaning. Our model captures these details through the encoding abilities of the hierarchical VQ-VAE. As a result, translations translations encompass emotional and emphatic cues beyond just words, significantly benefiting BLEU scores.

## 4.2 ABLATION STUDIES

Ablation analysis focuses on the parameter counts in pre-trained LLMs, the architectures of tokenizers, and RAG strategies (Table 2). These experiments involve selectively removing or modifying specific model features or structures to elucidate the impact of each component.

**Pre-trained Model Sizes.** We evaluate the performance across the four publicly accessible pre-trained models from FLAN-T5. The experimental results with the FLAN-T5-base show a compelling balance between size and performance (Table 2a). FLAN-T5-base achieve competitive accuracy in our tests, showing only a marginal decrease in performance compared to its larger counterparts. The FLAN-T5-base model excels in speed, showing an approximate 37% boost in inference speed, while the FLAN-T5-XL model surpasses in accuracy with a high of 36.38 in BLUE-1. The results make FLAN-T5-base an appealing choice for applications where efficiency is paramount. The slight trade-off in translation accuracy is more than offset by the gains in speed and resource efficiency, indicating that FLAN-T5-base is well-suited for resource-constrained environments.

**Sign-tokenizer.** To evaluate the impact of VQ-VAE structures, we experiment with VQ-VAE, VQ-VAE-2, and hierarchical VQ-VAE (Table 2b). The basic VQ-VAE model, although effective in encoding visual information, fell short in accurately translating complex gestures, achieving only a 34.08 BLEU-1. The improved VQ-VAE-2, with its more detailed encoding layers, raise BLEU-1 to 35.11. Further, our adoption of the hierarchical VQ-VAE, evolved from VQ-VAE-2, significantly enhances the capture of sign language details, boosting translation BLEU-1 to 35.61, thus proving its superiority in handling complex sign language information.

**Codebook Size.** In our pursuit to refine SLT accuracy, we vary the codebook sizes within sign-tokenizer and observe significant differences in ROUGE and BLEU scores (Table 2c). The size of codebook can directly influence the model's ability to quantize the input data. In general, larger embedding spaces can offer finer quantization. Initially, with a codebook size of 256, the model scored 27.94 in ROUGE and 30.66 in BLEU-1. Doubling the codebook size to 512 improved the ROUGE to 31.16 and BLEU to 34.47. However, when the embedding space reach a certain size (*i.e.*, 1024), performance improvement plateau, where the scores escalated to 33.38 for ROUGE and 35.61 for BLEU-1. These results underscore the importance of a larger codebook in capturing a broader array of features necessary for accurate translation. However, larger embedding spaces provide finer quantization but also introduce higher computational complexity and storage requirements. Thus, an code space size around 1024 offers a reasonable trade-off, providing good reconstruction performance while maintaining relatively low computational cost.

Table 2: Ablation studies on How2Sign (Duarte et al., 2021) dataset (§4.2).

**(a)** Pre-trained Model Size

| Methods | ROUGE↑ | BLEU-1↑ | BLEU-2↑ | BLEU-3↑ | BLEU-4↑ |
|---|---|---|---|---|---|
| FLAN-T5-small | $33.38_{\pm.06}$ | $35.61_{\pm.08}$ | $20.35_{\pm.12}$ | $13.12_{\pm.06}$ | $8.53_{\pm.04}$ |
| FLAN-T5-base | $34.06_{\pm.02}$ | $35.32_{\pm.04}$ | $21.17_{\pm.03}$ | $13.97_{\pm.06}$ | $9.20_{\pm.03}$ |
| FLAN-T5-large | $\mathbf{34.81}_{\pm.03}$ | $35.61_{\pm.04}$ | $21.54_{\pm.01}$ | $\mathbf{14.30}_{\pm.03}$ | $9.73_{\pm.05}$ |
| FLAN-T5-XL | $34.73_{\pm.01}$ | $\mathbf{36.38}_{\pm.04}$ | $\mathbf{21.80}_{\pm.03}$ | $13.76_{\pm.08}$ | $\mathbf{9.88}_{\pm.00}$ |

**(b)** Sign-tokenizer

| Methods | ROUGE↑ | BLEU-1↑ | BLEU-2↑ | BLEU-3↑ | BLEU-4↑ |
|---|---|---|---|---|---|
| VQ-VAE | $30.80_{\pm.04}$ | $34.08_{\pm.05}$ | $18.16_{\pm.02}$ | $10.01_{\pm.03}$ | $8.04_{\pm.07}$ |
| VQ-VAE-2 | $32.47_{\pm.02}$ | $35.11_{\pm.06}$ | $19.48_{\pm.04}$ | $12.92_{\pm.05}$ | $8.29_{\pm.05}$ |
| Sign-tokenizer | $\mathbf{33.38}_{\pm.02}$ | $\mathbf{35.61}_{\pm.04}$ | $\mathbf{20.35}_{\pm.03}$ | $\mathbf{13.12}_{\pm.06}$ | $\mathbf{8.53}_{\pm.03}$ |

**(c)** Codebook Size

| Methods | ROUGE↑ | BLEU-1↑ | BLEU-2↑ | BLEU-3↑ | BLEU-4↑ |
|---|---|---|---|---|---|
| Sign-tokenizer-128 | $27.94_{\pm.08}$ | $30.66_{\pm.05}$ | $15.39_{\pm.09}$ | $10.68_{\pm.02}$ | $6.13_{\pm.07}$ |
| Sign-tokenizer-256 | $31.16_{\pm.03}$ | $34.47_{\pm.04}$ | $19.61_{\pm.02}$ | $11.81_{\pm.01}$ | $7.42_{\pm.07}$ |
| Sign-tokenizer-512 | $\mathbf{34.84}_{\pm.02}$ | $34.65_{\pm.04}$ | $\mathbf{20.93}_{\pm.08}$ | $12.60_{\pm.05}$ | $8.33_{\pm.13}$ |
| Sign-tokenizer-1024 | $33.38_{\pm.02}$ | $\mathbf{35.61}_{\pm.04}$ | $20.35_{\pm.03}$ | $\mathbf{13.12}_{\pm.06}$ | $\mathbf{8.53}_{\pm.03}$ |

**(d)** RAG

| Methods | ROUGE↑ | BLEU-1↑ | BLEU-2↑ | BLEU-3↑ | BLEU-4↑ |
|---|---|---|---|---|---|
| *w/o* RAG | $32.16_{\pm.05}$ | $33.41_{\pm.08}$ | $19.39_{\pm.05}$ | $12.83_{\pm.04}$ | $7.63_{\pm.06}$ |
| Pre-SignLLM | $33.06_{\pm.06}$ | $35.32_{\pm.03}$ | $\mathbf{20.87}_{\pm.02}$ | $12.97_{\pm.06}$ | $8.20_{\pm.04}$ |
| Post-SignLLM | $\mathbf{33.38}_{\pm.02}$ | $\mathbf{35.61}_{\pm.04}$ | $20.35_{\pm.03}$ | $\mathbf{13.12}_{\pm.06}$ | $\mathbf{8.53}_{\pm.03}$ |

**RAG.** We explore three RAG configurations (Table 2d): no RAG, RAG applied before the LLM (pre-SignLLM), and RAG applied after the LLM (post-SignLLM). Our findings indicate significant differences in translation accuracy across these setups. Without retrieval enhancement, RAG-SLT relies on pre-trained knowledge, leading to insufficient handling of new information. Without RAG, VRG-SLT achieves a BLEU-1 score of 54.90 and a ROUGE score of 52.25. Pre-SignLLM results in improved performance, with the BLEU score rising to 54.83 and the ROUGE score to 54.26. Moreover, post-SignLLM yields the best results, with a BLEU score of 53.92 and a ROUGE score of 55.74. RAG can enhance the knowledge coverage by retrieving from external knowledge bases, which is particularly useful for generating knowledge-based answers.

## 4.3 DISCUSSION

**Impacts.** Sign language translation technology bridges communication gaps, providing the deaf and hard-of-hearing community with greater access to information and services. Socially, this fosters inclusivity, ensuring that individuals who use sign language can participate fully in educational, professional, and social settings. This technology can empower deaf communities by providing more autonomous and straightforward ways to communicate, reducing reliance on interpreters. From a technological standpoint, advancements in this field drive innovation in NLP and computer vision, pushing the boundaries of how machines understand and interpret human gestures and expressions.

**Limitation.** VRG-SLT still struggles with the contextual and cultural nuances of sign languages. Sign languages are not universal and vary widely from one region to another. Thus, its often fail to account for these variations, leading to translations that may be correct in one dialect but completely off in another. This lack of sensitivity to regional differences can significantly affect the utility of translation technologies. Additionally, the subtleties of hand shapes, orientations, and movements in sign language can be difficult to capture reliably, especially in complex or dynamic environments. This limitation often results in errors or inaccuracies in translation, hindering effective communication.

## 5 CONCLUSION

We present VRG-SLT as a unified framework for sign language translation, generating spoken descriptions based on prompt-driven instructions. Extensive experiments on How2Sign and PHOENIX-2014T datasets demonstrate competitive performance and validate the efficacy of each module. The hierarchical VQ-VAE effectively encodes visual gestures into a compressed representation, playing a vital role. Simultaneously, SignLLM establishes a robust linguistic framework that enhances translation with a deep understanding of syntax and semantics. Collectively, these components push the boundaries of traditional SLT methods, achieving a BLEU-1 score improvement from 53.97 to 55.74 and a ROUGE score from 52.65 to 53.92. The collaborative training of VQ-VAE and LLMs offers promising tools for nuanced communication within the deaf community, showcasing the transformative effects on accessibility and interaction. Future research aspires to break down linguistic boundaries, enabling multilingual translation within a unified model.

**Ethics Statement**. The generative power of our model stems from the large language model FLAN-T5, which has been fine-tuned to include extra knowledge relevant to sign language. Our model also shares ethical and legal considerations with FLAN-T5. We employ open-source sign language datasets and knowledge bases that adhere to applicable ethical norms and laws. There are no human subjects involved in our experimental processes. Large language models will not scrutinize the sign tokens entered into the system. Instead, they attempt to generate output based on the received token sequence, significantly influenced by the sign-tokenizer.

**Reproducibility Statement.** In the main text, we highlight the fundamental techniques for building our framework in the first stage (§3.1), second stage (§3.2), and third stage (§3.3). Our experimental data are drawn from widely-used public datasets, and training steps are discussed in §3.3. Detailed model configurations, all optimizer hyperparameters, and model dimensions are elaborated in §A. Additionally, pseudocode for each stage is provided in §B.

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

# SUPPLEMENTARY MATERIALS

This supplementary material, provided for a more comprehensive understanding of the main paper, is organized as follows:

- § A: **Architecture Details**. Detail the VRG-SLT network architecture, including its layer composition and connectivity patterns, *etc*.
- § B: **Pseudo code**. Outline the execution steps for each stage.
- § C: **Competitors**. Provide a brief overview of the methods compared in experiments.
- § D: **More Experimental Analysis**.

## A   ARCHITECTURE DETAILS

VRG-SLT consists of three key components: the sign-tokenizer, the large language model FLAN-T5, and the RAG. Among these, FLAN-T5 utilizes the pre-trained Base version with approximately 220 million parameters. RAG is implemented using BERT. The sign-tokenizer, inspired by the design of motionGPT, mainly employs 1x1 convolutional neural networks. Detailed network settings are provided in Table A2.

Table A1: Network Configuration Details

| Model | Flan-T5-Base |
|---|---|
| Training Batch Size | 16 |
| Model Size | $220M$ |
| Pre-training - Iterations | $300K$ |
| Pre-training - Learning Rate | $2e-4$ |
| Instruction Tuning - Iterations | $300K$ |
| Instruction Tuning - Learning Rate | $1e-4$ |

Table A2: Network Configuration Details

| Module | Encoder $\mathcal{E}_u$ | Encoder $\mathcal{E}_h$ | Decoder $\mathcal{D}_u$ | Decoder $\mathcal{D}_h$ |
|---|---|---|---|---|
| Conv1d | 1 | 1 | 1 | 1 |
| Resnet1D-block | 5 | 5 | 5 | 5 |
| Conv1d | 1 | 1 | 1 | 1 |
| Vocabulary Number $V_m$ | 1024 | | | |
| Codebook Dimension | 512 | | | |
| Batch Size | 512 | | | |
| Iterations | $150k$ | | | |
| Learning Rate | $1e-4$ | | | |

Human keypoints are employed in training the sign-tokenizer, which brings several benefits to improving the precision and efficiency of sign language translation:

- **Precise Hand Localization:** Utilizing human body keypoints allows for direct and precise extraction of hand regions, crucial for capturing subtle gestures in sign language.

- **Focusing on Relevant Features:** Keypoints concentrate on critical aspects of sign language, such as hand positions, facial expressions, and body postures. This focus allows the sign-tokenizer to capture the essential elements of sign language more accurately without being distracted by background noise or irrelevant details.

- **Robustness to Variability:** Normalization of keypoints enhances the model's robustness against variations in environment, camera distances, and different lighting conditions.

- **Efficiency in Processing:** Compared to processing full RGB video frames, keypoints effectively reduce computational load. By simplifying gestures into a more basic form, they streamline the processing and can speed up recognition and translation tasks.

# B PSEUDO CODE

Our sign language translation framework, VRG-SLT, comprises three training stages. This section provides pseudocode for each stage.

---

**Stage 1:** Trainging of sign-tokenizer

---

```
# Initialize the hierarchical encoders, decoders, and quantizers
def training(sign_motions, hand_motions):
    # Encode images at multiple scales to get hierarchical latent representations
    Z_upper = encode_upper(sign_motions); Z_hand = encode_hand(hand_motions)

    # Quantize the latent representations of upper body
    Q_upper = vector_quantize_top(Z_upper)
    Dec_upper = decoder_upper(Q_upper)

    # Combine quantized representations from different scales
    combined_Z = combine(Dec_upper, Z_hand)
    Q_hand = vector_quantize_hand(combined_Z)

    # Decode combined quantized representations to reconstruct motions
    reconstructed_motions = decode(Q_hand)

    # Compute reconstruction loss between original motions and reconstructed_motions
    reconstruction_loss = compute_loss(sign_motions, reconstructed_motions)

    # Compute quantization loss for top and bottom levels
    quantization_loss_upper = compute_loss(Z_upper, Q_upper)
    quantization_loss_hand = compute_loss(Z_hand, Q_hand)

    # Total quantization loss is the sum of top and bottom quantization losses
    total_quantization_loss = quantization_loss_upper + quantization_loss_hand

    # Total loss is the sum of reconstruction loss and total quantization loss
    total_loss = reconstruction_loss + total_quantization_loss

    # Update model parameters based on total_loss
    update_parameters(total_loss)
```

---

---

**Stage 2:** Finetuning the large language model FLAN-T5

---

```
# Initialize the FLAN-T5 model
def finetuning(batch):
    # Inputs are sign language tokens combined with text prompts
    sign_tokens, text_prompts = batch['sign_tokens'], batch['text_prompts']

    # Combine tokens with prompts to form the input for the model
    model_input = concatenate(sign_tokens, text_prompts)

    # Expected translation as output
    expected_output = batch['translated_text']

    # Perform model training with input and expected output
    loss = train_model(model_input, expected_output)

    # Update model parameters based on the loss
    update_parameters(loss)

def inference(sign_tokens, text_prompt):
    # Combine sign language tokens with text prompt for inference
    input_for_inference = concatenate(sign_tokens, text_prompt)

    # Generate translation using the fine-tuned model
    translated_text = generate_translation(input_for_inference)

    return translated_text

# Example sign_tokens and text_prompt for testing inference
test_sign_tokens = ['sign_token1', 'sign_token2', 'sign_token3']
test_text_prompt = "Translate_the_following_sign_language_sequence:"
translation = inference(test_sign_tokens, test_text_prompt)
print("Translated_text:", translation)
```

---

**Stage 3:** Tuning with RAG

```
# Initialize the BERT model for retrieval and a RAG module for refinement
def indexing(corpus):
    # Index the corpus with BERT to facilitate efficient retrieval
    indexed_corpus = bert_index(corpus)
    return indexed_corpus

def retrieving(initial_translation, indexed_corpus):
    # Use BERT to retrieve relevant documents or context from the indexed corpus
    retrieved_documents = bert_retrieve(initial_translation, indexed_corpus)
    return retrieved_documents

def generation(initial_translation, retrieved_context):
    # Combine the initial translation from FLAN-T5 with retrieved context
    combined_input = concatenate(initial_translation, retrieved_context)

    # Use the generator model to refine the translation
    refined_translation = generator_model(combined_input)
    return refined_translation

def inference(sign_tokens, text_prompt):
    # Generate initial translation using FLAN-T5
    input_for_inference = concatenate(sign_tokens, text_prompt)
    initial_translation = flan_t5_generate_translation(input_for_inference)

    # Index the relevant corpus if not already indexed (can be pre-indexed)
    indexed_corpus = index_corpus(corpus) # Assuming 'corpus' is predefined or loaded

    # Retrieve context based on the initial translation
    context = retrieve_context(initial_translation, indexed_corpus)

    # Generate the final, refined translation using the retrieved context
    final_translation = generate_refined_translation(initial_translation, context)
    return final_translation

# Example usage:

# Assuming corpus is available and FLAN-T5 is pre-trained
test_sign_tokens = ['sign_token1', 'sign_token2', 'sign_token3']
test_text_prompt = "Translate the following sign language sequence:"
final_translation = inference(test_sign_tokens, test_text_prompt)
print("Final Translated text:", final_translation)
```

## C  COMPETITORS

We offer succinct introductions to a few state-of-the-art methods compared in this paper:

- **SL-Luong** (Camgöz et al., 2018) distinguishes sign language translation from traditional sign language recognition by addressing it as a complex translation problem. By framing SLT in pretrained contexts, it effectively captures spatial representations and the intricate mapping between sign and spoken languages, acknowledging the unique grammatical structures of sign languages.

- **TSPNet-Joint** (Li et al., 2020a) is a temporal semantic pyramid network that innovatively learns hierarchical sign video features without precise segmentation. The network employs a new segment representation and attention mechanisms at multiple scales to improve the accuracy and consistency.

- **SL-Transf** (Camgöz et al., 2020) integrates continuous sign language recognition and translation using CTC loss. This method obviates the need for ground-truth timing and significantly boosts performance by solving interdependent learning challenges concurrently.

- **STMC-T** (Zhou et al., 2022) incorporates multi-cue learning into neural networks to capture the nuanced visual grammars of sign language. It consists of spatial and temporal modules that separately and jointly analyze visual cues, achieving end-to-end sequence learning.

- **TIN-Trans** (Zhou et al., 2021) introduces a sign back-translation strategy to mitigate the parallel data bottleneck in SLT. It back-translates text to gloss and then assembles sign sequences from a gloss bank, thus enriching the training dataset for the SLT encoder-decoder framework.

- **SLRT** (Chen et al., 2022) is a transfer learning approach for sign language translation, addressing the data scarcity issue by progressively pretraining on general and specific domain datasets. It includes pretraining separate networks for sign-to-gloss and gloss-to-text translations, which are then connected by a visual-language mapper for fine-tuning.

- **SLTUNET** (Zhang et al., 2023a), a unified neural model, is proposed to support various sign language translation tasks, effectively bridging the modality gap and mitigating data scarcity issues. The model explores cross-task relatedness and taps into external spoken language data.
- **SIGN2GPT** (Hu et al., 2023) merges computer vision and language processing, using lightweight adapters with large pretrained models to overcome data scarcity. It leverages pseudo-glosses to train the encoder, eliminating the need for precise gloss annotations.
- **SignBERT+** (Wong et al., 2024) is a self-supervised framework designed to improve sign language understanding by integrating a hand prior aware of model contexts. Hand gestures are encoded as visual tokens with detailed position and gesture information.

## D    MORE EXPERIMENTAL ANALYSIS

**Ablation studies on PHOENIX-2014T**. To further elucidate the impact of different components on our sign language translation model, we conducted additional ablation studies using the PHOENIX-2014T dataset (Table A3). This section aims at dissecting the contribution of each component to the overall performance of our translation model.

As shown in Table A3, similar outcomes are present within the How2Sign dataset. For instance, in the sign-tokenizer experiments, our sign-tokenizer consistently outperforms other methods, due to better capture of sign language nuances or more effective learning strategies. The gradual decrease in BLEU scores from BLEU-1 to BLEU-4 across all methods indicates that generating longer coherent text sequences remains a challenge.

Similarly, The addition of RAG components (Pre and Post SignLLM) generally improves performance over the baseline, underscoring the value of incorporating retrieval-augmented strategies in handling complex language tasks like sign language translation. *w/o* RAG represents the baseline model without the Retrieval-Augmented component, showing robust initial scores but lower in more complex metric evaluations (BLEU-3 and BLEU-4). Pre-SignLLM shows improvement over the baseline, particularly in ROUGE and BLEU-1 scores, suggesting that pre-processing or prior learning can enhance performance. Post-SignLLM is similar to Pre-SignLLM in ROUGE, but slightly better in higher BLEU metrics, implying further enhancements post-initial training. Consistent with the sign-tokenizer results, there is a noticeable performance drop in higher BLEU metrics, indicating the inherent difficulty of the tasks as they require maintaining longer-range textual coherence.

Table A3: More ablation studies on PHOENIX-2014T (Camgöz et al., 2018) dataset (§4.2).

**(a) Pre-trained Model Size**

| Methods | ROUGE↑ | BLEU-1↑ | BLEU-2↑ | BLEU-3↑ | BLEU-4↑ |
|---|---|---|---|---|---|
| FLAN-T5-small | $53.71_{\pm.04}$ | $54.92_{\pm.06}$ | $42.84_{\pm.01}$ | $36.22_{\pm.03}$ | $28.36_{\pm.04}$ |
| FLAN-T5-base | $53.92_{\pm.05}$ | $\mathbf{55.74}_{\pm.01}$ | $43.31_{\pm.01}$ | $36.59_{\pm.05}$ | $30.17_{\pm.06}$ |
| FLAN-T5-large | $55.80_{\pm.01}$ | $55.39_{\pm.06}$ | $43.04_{\pm.07}$ | $38.52_{\pm.05}$ | $32.44_{\pm.02}$ |
| FLAN-T5-XL | $\mathbf{56.59}_{\pm.04}$ | $55.05_{\pm.04}$ | $\mathbf{44.17}_{\pm.00}$ | $\mathbf{38.94}_{\pm.05}$ | $\mathbf{32.83}_{\pm.03}$ |

**(b) Sign-tokenizer**

| Methods | ROUGE↑ | BLEU-1↑ | BLEU-2↑ | BLEU-3↑ | BLEU-4↑ |
|---|---|---|---|---|---|
| VQ-VAE | $50.10_{\pm.08}$ | $52.93_{\pm.05}$ | $41.96_{\pm.02}$ | $34.52_{\pm.01}$ | $28.61_{\pm.10}$ |
| VQ-VAE-2 | $53.05_{\pm.05}$ | $53.62_{\pm.03}$ | $43.13_{\pm.01}$ | $35.23_{\pm.07}$ | $29.58_{\pm.02}$ |
| Sign-tokenizer | $\mathbf{53.92}_{\pm.05}$ | $\mathbf{55.74}_{\pm.01}$ | $\mathbf{43.31}_{\pm.01}$ | $\mathbf{36.59}_{\pm.05}$ | $\mathbf{30.17}_{\pm.06}$ |

**(c) Codebook Size**

| Methods | ROUGE↑ | BLEU-1↑ | BLEU-2↑ | BLEU-3↑ | BLEU-4↑ |
|---|---|---|---|---|---|
| Sign-tokenizer-128 | $48.27_{\pm.11}$ | $50.64_{\pm.06}$ | $36.87_{\pm.06}$ | $33.39_{\pm.03}$ | $26.81_{\pm.09}$ |
| Sign-tokenizer-256 | $50.53_{\pm.08}$ | $52.19_{\pm.02}$ | $39.53_{\pm.02}$ | $34.84_{\pm.04}$ | $26.25_{\pm.07}$ |
| Sign-tokenizer-512 | $52.36_{\pm.04}$ | $53.27_{\pm.01}$ | $41.94_{\pm.01}$ | $35.26_{\pm.03}$ | $29.30_{\pm.06}$ |
| Sign-tokenizer-1024 | $\mathbf{53.92}_{\pm.05}$ | $\mathbf{55.74}_{\pm.01}$ | $\mathbf{43.31}_{\pm.01}$ | $\mathbf{36.59}_{\pm.05}$ | $\mathbf{30.17}_{\pm.06}$ |

**(d) RAG**

| Methods | ROUGE↑ | BLEU-1↑ | BLEU-2↑ | BLEU-3↑ | BLEU-4↑ |
|---|---|---|---|---|---|
| *w/o* RAG | $52.25_{\pm.06}$ | $54.90_{\pm.02}$ | $42.70_{\pm.07}$ | $35.11_{\pm.10}$ | $29.49_{\pm.01}$ |
| Pre-SignLLM | $\mathbf{54.83}_{\pm.03}$ | $54.26_{\pm.10}$ | $42.97_{\pm.06}$ | $36.08_{\pm.02}$ | $29.70_{\pm.04}$ |
| Post-SignLLM | $53.92_{\pm.05}$ | $\mathbf{55.74}_{\pm.01}$ | $\mathbf{43.31}_{\pm.01}$ | $\mathbf{36.59}_{\pm.05}$ | $\mathbf{30.17}_{\pm.06}$ |

