# OpenReview forum: "HYBRID MODEL COLLABORATION FOR SIGN LANGUAGE TRANSLATION WITH VQ-VAE AND RAG ENHANCED LLMS"
_ICLR.cc/2025/Conference — ICLR 2025 Conference Withdrawn Submission_

### Official Review · Reviewer_RLDc · 2024-10-29

**Soundness:** 3
**Presentation:** 3
**Contribution:** 2
**Rating:** 6
**Confidence:** 3

**Summary:**

The paper presents VRG-SLT, an innovative framework for translating sign language into spoken language, aiming to facilitate communication between signing and non-signing communities. The key contributions of the paper are:

VRG-SLT Framework: The authors introduce a two-stage pipeline that utilizes a hierarchical VQ-VAE (Vector Quantized Variational Autoencoder) to convert continuous sign sequences into discrete representations called sign codes. These codes are then aligned with text using a fine-tuned pre-trained language model (FLAN-T5). The framework also employs a Retrieval-Augmented Generation (RAG) strategy to enhance the language model's output with relevant factual knowledge, improving the accuracy and coherence of the generated text.

Sign-tokenizer: A novel component that captures both overall upper body and hand trajectory characteristics using a hierarchical structure. This allows for a comprehensive understanding of sign language gestures.

Integration of RAG: The incorporation of RAG enables the retrieval and combination of relevant knowledge to refine the initial translations, addressing issues like "hallucination" (producing inaccurate or unrealistic responses) in large language models.

**Strengths:**

Innovative Framework: although VQVAE has been widely adopted in computer vision area, the application to sign laugnage translation is a contribution of the paper. Furthermore, it integrats a hierarchical VQ-VAE with a pre-trained language model (FLAN-T5) and a Retrieval-Augmented Generation (RAG) strategy, the framework addresses the complexities of translating sign language into spoken text effectively.

Well-Written: readers could easily follow the story and understand how to reproduce the paper.

**Weaknesses:**

1. While the paper acknowledges the challenge of data scarcity in sign language, it primarily focuses on developing a model to mitigate this issue rather than addressing the issue from data perspective. I admit that the model improvement will sightly benefit the end-to-end performance, but I believe the final solution for the task is data augmentation or self-supervised learning.

Looking back at the history of machine translation, the most effective methods have been back translation, pivot-translation (data augmentation), and self-supervised learning (GPT series models). While various methods have been proposed to address data sparsity from a modeling perspective, they remain theoretical solutions rather than definitive answers to the problem or adopted by industry.

2. When we look at the performance, the paper misses some recent baselines (e.g. https://arxiv.org/pdf/2211.01367) and the improvement over these baselines are limited.

Overall, although the paper is well-written, it is unlikely to revolutionize sign language translation.

**Questions:**

When using RAG (Retrieval-Augmented Generation) in the framework, can you guarantee that the retrieved content maintains the original meaning of the sentences?  I think  the retrieved content might be semantically similar but not exactly matching the intended meaning, leading to subtle shifts in meaning during generation.

---

> ### Author Response · Authors · 2024-11-26
>
> We sincerely thank the reviewer for the time and constructive feedback. We address the main concerns below.
>
> **Q1. Discussion on the definitive solutions for overcoming data scarcity in sign language translation tasks: data augmentation or self-supervised learning.**
>
> **A:** Thank you very much for your valuable comments. We strongly agree with your comment that data augmentation or self-supervised learning is the ultimate solution. Due to the small number of sign language users, especially in countries with very small populations, sign language data faces a serious challenge of insufficient data. Therefore, we use the rich prior knowledge embedded in the large model to alleviate the challenge of insufficient data, and use the understanding of world knowledge and inherent multi-domain knowledge in the large model to achieve knowledge sharing and transfer.
>
> **Q2.  Discussion on subtle changes in meaning during text generation with the RAG model.**
>
> **A:** Thank you for pointing this out. Subtle changes in meaning can indeed impact the accuracy of information retrieval and the quality of generated outputs in RAG, leading to issues such as context misalignment, reduced coherence, knowledge gaps, and lowered retrieval precision. To mitigate these effects, we employ a **Top-3 retrieval** strategy (*i.e.*, selecting the top K most relevant results), ensuring that the system has access to more information during generation  (**Lines 353-355**). This approach helps avoid over-reliance on a single document and better addresses subtle semantic variations.

---

### Official Review · Reviewer_x9jx · 2024-10-31

**Soundness:** 2
**Presentation:** 2
**Contribution:** 2
**Rating:** 3
**Confidence:** 4

**Summary:**

The paper studies sign language translation (sign --> text) with VQ-VAE sign tokenization adapted into a pretrained encoder-decoder language model (Flan T5), on How2Sign and PHOENIX 2014T.

**Strengths:**

* The method in the paper is presented relatively clearly.
* The experiments look reasonable.

**Weaknesses:**

At a high level, primarily I am skeptical of the novelty/contribution of the work, and secondarily some aspects of the paper seem unjustified.

* There is a lot of missing related work that undermines the contribution of the paper. For example:
  - There are many prior works that use VQ-VAE for sign language. (e.g., https://arxiv.org/abs/2404.00925 (this one especially), https://aclanthology.org/2024.signlang-1.24/, https://arxiv.org/abs/2208.09141v3)
  - There are many prior works that use "LLMs", even T5, as the pretrained base for sign language translation. (e.g., https://arxiv.org/abs/2306.15162, https://aclanthology.org/2024.acl-long.467/ (which you cited but not recognizing this), https://arxiv.org/abs/2406.06907, https://arxiv.org/abs/2405.04164) This is already a pretty dominating trend in sign language translation research in the last year or two.
* At the same time, I think there is a lot of unnecessary related work, like you don't need to go into details about the history of SLR or pre-Camgoz et al 2018 sign language translation.
* More broadly, the description of contributions in the paper is muddy. On lines 122-131, I would say that (1) and (2) are not novel contributions. I've given you pointers above to prior works that already train the sign language modality into LLMs, and use VQ-VAEs. It's possible there are novel elements to your work in these regards, but they are not articulated here. (3) is I suppose novel but I have more comments on that below.
* I don't see it articulated anywhere why we should prefer discrete tokens as input to an encoder-decoder translation model, rather than soft tokens (which don't impose an information bottleneck), as in many other sign language translation papers (and many multimodal fusion in LLM papers).
* I don't understand the point of RAG here. Retrieval-augmented translation typically retrieves from parallel translation datasets as a way to improve model capacity. RAG instead from knowledge bases is intended to improve factuality, but translation is not a factual task: you can translate sentences that are false. So sure, you could maybe get improvement on sentences that are factual, or when world knowledge might help you guess words by context, but this fails on nonfactual content. Also, how do you know the RAG is even being used, and it isn't just the LLM doing rewriting (or integrating its own knowledge)? It doesn't seem like you ablate LLM rewriting without any retrieved sentences, but I could be misunderstanding.
* I won't count it against you too much since maybe you don't have the resources to use larger datasets or can't use the datasets for license reasons, but there are much larger sign language datasets available (YouTube-ASL, BOBSL) and it is not clear how meaningful results on smaller datasets are. For example, the main downside of using VQ-VAE as an input representation is that you are imposing an information bottleneck, which may be irrelevant when training data is small enough that no results are particularly good.
* I am a bit confused by Table 1. You are excluding numerous works that score better on How2Sign, which I am guessing is because you are trying to compare to works that train on the same amount of sign language data. But it seems misleading to do this and call it "state-of-the-art" without explaining this, especially because SLTUNet trains on extra sign language gloss data, LLMs might train on sign language gloss data and explanations on the web, etc; this doesn't seem like a principled distinction. Also: where are these How2Sign numbers for prior works coming from? I looked at e.g. SignBERT+ and SLTUNet and neither of the works evaluates on How2Sign. Did you get these from personal correspondence with the authors? Did you reproduce all of these using their methods? The PHOENIX numbers I can see in the original papers.
* There are a bunch of weird claims throughout the paper that aren't justified. For example:
  - 48-49: "These differences, especially in word order, make transcription between the two languages complex": I think you mean "translation" here, not "transcription", but regardless I wouldn't say word order is a primary reason that translation between spoken languages and sign languages is hard. Different spoken languages have different word order and it isn't really an issue. The unique aspects of sign language grammar are related to use of space, non-manual markers, etc. The paper doesn't engage with this anyway, just the multimodal fusion.
  - 233-235: "The two encoders of the sign-tokenizer encode global body movements and detailed hand features, respectively, achieving a comprehensive and precise understanding of sign motion.": I see no evidence in the paper that the tokenizer encodes sign motion comprehensively or precisely. This could be proven by, for example, getting human ratings of the reconstructions from fluent sign language users.
  - 520: "Limitations: VRG-SLT still struggles with the contextual and cultural nuances of sign languages." This is a massive understatement; models that get ~8 BLEU on How2Sign are extremely poor quality in absolute. They are nowhere close to context or cultural nuances being the main limitations.

**Questions:**

These are essentially written in Weaknesses above, but some additional questions:

* Could you provide examples of outputs before and after "RAG" LLM rewriting? Like when is a relevant sentence ever retrieved from the ECMWF weather dataset? It seems like this could only possibly help if your translation draft is something like "On September 12, 2007, the weather in Cologne was [X]" and your knowledge base has memorized the weather.
* Could you elaborate on what "w/o RAG", " pre-SignLLM", and "post-SignLLM" mean precisely in Table 2d?
* Figures 2, 3, and 4 are misleading in that they show the sign language input to the encoder model as image frames. But line 407 says that they are actually keypoints (derived from what model?).

---

> ### Author Response · Authors · 2024-11-26
>
> We thank the reviewer for the time and constructive feedback. We address the main questions below.
>
> **Q1. Skepticism about novelty and contribution.**
>
> **A:** We disagree. Our approach is novel and pioneering. It is unreasonable and odd to negate the novelty simply because previous work has used LLMs and VQ-VAE. To the best of our knowledge, we are the first to apply multi-level VQ-VAE and FLAN-T5 to a gloss-free sign language translation task. Furthermore, with the integration of RAG, we utilize a knowledge base to improve sign language-to-text translation, making us the first to leverage RAG for enhancing translation performance.
>
> The reviewer points out that other methods also employ LLMs and emphasizes [ref1]. Our method, however, differs substantially from [ref1], mainly in the following aspects: (1) [Ref1] focuses on the discrete features and hierarchical structure of symbolic tokens, normalizing sign sentences to reflect two core linguistic properties: discreteness and hierarchy. In contrast, our sign tokenizer focuses on the hierarchical reconstruction of sign actions. (2) [Ref1] freezes the off-the-shelf LLM to retain the rich knowledge acquired during pretraining, while we fine-tune the LLM to better align with the sign tokens produced by the sign tokenizer and preserve prior knowledge. Additionally, our method outperforms [ref1] on the standardized Phoenix-2014T dataset in terms of evaluation metrics, as shown in the following table. In summary, both the framework design and experimental results demonstrate the novelty and contribution of our approach.
>
> | Methods | ROUGE↑| BLEU-1↑| BLEU-2↑| BLEU-3↑ | BLEU-4↑|
> |----------|----------|----------|----------|----------|----------|
> | [ref1] | 44.49 | 45.21| 34.78| 28.05| 23.40 |
> | Ours | 53.92| 55.74| 43.31| 36.59| 30.17 |
>
> [ref1] LLMs are Good Sign Language Translators. CVPR 2024.
>
> **Q2. Lack of analysis and discussion concerning some relevant studies.**
>
> **A:**  Thank you for the references you provided. We have already compared some of the papers you mentioned [ref5]. However, most of the references [ref2,3,4] you mentioned are irrelevant to our task, especially [ref3], which addresses the inverse task of our sign language translation.
>
> As for T5 [ref6], there is no significant difference in network architecture compared to FLAN-T5 [ref7]; the main difference lies in the training strategy and task fine-tuning. We also mention the reason for using FLAN-T5 in our manuscript (**Lines 191-193, 316-318).** In addition, we also compared different parameter versions of FLAN-T5 in the ablation experiments. Therefore, our results are convincing.
>
> [ref2] Exploring Latent Sign Language Representations with Isolated Signs, Sentences and In-the-Wild Data. ACL 2024.
>
> [ref3] G2P-DDM: Generating Sign Pose Sequence from Gloss Sequence with Discrete Diffusion Model. AAAI 2024.
>
> [ref4] Towards Privacy-Aware Sign Language Translation at Scale. ACL 2024.
>
> [ref5] Sign2GPT: Leveraging Large Language Models for Gloss-Free Sign Language Translation. ICLR 2024.
>
> [ref6] Exploring the limits of transfer learning with a unified text-to-text transformer. JMLR, 2020.
>
> [ref7] Scaling Instruction-Finetuned Language Models. JMLR, 2024.
>
>
> **Q3. Unnecessary related work, such as a detailed history of SLR or sign language translation prior to 2018.**
>
> **A:** We do not share this view. First, we do not provide a detailed history of SLR. Instead, we outlined the development of sign language translation and emphasized areas that could lead to misunderstandings between different tasks. SLR is a fundamental task in sign language understanding and is once considered synonymous with sign language translation. Historically, sign language translation typically involves two steps: SLR and gloss-to-text [ref1,2,5]. Including an overview of SLR helps contextualize the background of sign language translation and clarify the differences between task settings. Therefore, we believe it is essential to mention some SLR-related work in the related works section of our paper.
>
> **Q4. The paper's description of its contributions is muddy.**
>
> **A:** We respectfully disagree. In Introduction section, we clearly describe the motivation behind our method and its uniqueness. Moreover, in **Q1** and **Q2**, we clearly reiterated the novelty of our approach and emphasized its differences from [ref1].

---

> > ### Author Response · Authors · 2024-11-26
> >
> > **Q5.  Discussion on the role of RAG within the overall framework and its effectiveness.**
> >
> > **A:** In this paper, RAG uses the pre-trained large-scale model BERT for information retrieval to improve the quality and accuracy of sign language translation. The key idea is that, during translation, the model not only relies on its internal knowledge (gained through pre-training) but also retrieves relevant information from external knowledge bases to refine and enhance the translation (**Lines 101-102, 113-119**). The initial translation obtained from the large model is refined by RAG based on retrieved information. Note that our ablation experiments did not compare the performance with and without RAG. The results demonstrate that the RAG module positively impacts translation performance.
> >
> > **Q6. Discussion on the limited significance of experimental results from smaller datasets.**
> >
> > **A:** We disagree with this point. As you mentioned in the previous question, most studies [ref1,5] focus on the PHOENIX-2014T dataset, leading to a lack of results for How2Sign. Moreover, How2Sign is currently one of the largest datasets in sign language research and is increasingly being adopted as a standard for sign language translation evaluation. Thus, these two datasets provide ample support for assessing the effectiveness of our approach.
> >
> > **Q7. Reasons for method selection in Table 1 and the sources of How2Sign dataset results.**
> >
> > **A:** Table 1 presents a comparison of several classic SOTA sign language translation methods with different pipelines. The results on the How2Sign dataset are derived by implementing these methods based on their open-source code and paper details, with experimental setups and dataset handling aligned with our approach.
> >
> > **Q8.  Explanation of how differences, particularly in word order, complicate transcription between the two languages as discussed in Lines 48-49.**
> >
> > **A:** I beg to differ.  I would like to clarify that what I meant in this sentence is that differences in word order make translation between the two languages more complex. It is not as you claimed, that I said word order is the primary reason for the difficulty in translating between spoken and sign languages. 'Becoming complex' and 'primary reason' are not the same thing. Moreover,  word order is widely acknowledged in the papers as a challenge in translating between spoken and sign languages. This has become a consensus in the field of sign language translation, particularly in the reference you mentioned. Grammatical differences between sign language and spoken language complicate translation. For example, while English typically follows a subject-verb-object structure, American Sign Language (ASL) may use subject-object-verb or other patterns. This requires not only literal translation but also adjustments to align with the target language's grammar. Moreover, sign language conveys emotional and contextual nuances through gestures, facial expressions, and body language, which often require more detailed descriptions in spoken languages. As a result, translating sign language involves both grammatical conversion and capturing its subtle nuances, adding complexity to the process.
> >
> > **Q9. The explanation of the word ''transcription.''**
> >
> > **A:** **Transcription** is defined in the Oxford English Dictionary as ''the action or process of transcribing something'' and ''a written or printed version of something,'' such as text derived from speech or signs. In the field of sign linguistics, ''sign language transcription'' is a widely used term to describe the process of converting sign language actions or symbols into written form. It is reasonable for us to use ''transcription'' interchangeably with ''translation'' in our paper.

---

> > > ### Author Response · Authors · 2024-11-26
> > >
> > > **Q10.  Discussion on whether the tokenizer can accurately encode sign language gestures and the manual evaluation of reconstruction by fluent sign language users (Lines 233-235).**
> > >
> > > **A:** We find it difficult to agree. First, recruiting enough fluent users of American Sign Language and German Sign Language is both challenging and unrealistic. Second, VQ-VAE is a mature, unsupervised method that has been successfully validated in almost all deep learning fields, designed to replicate the original output. Finally, our implementation is primarily based on [ref8], and VQ-VAE reconstruction visualizations are also provided in [ref8]. Accordingly, we will also provide the code for reconstruction visualizations.
> > >
> > > [ref8] MotionGPT: Human Motion as a Foreign Language. NeurIPS 2023.
> > >
> > > **Q11.  Explanation of the challenges faced by VRG-SLT regarding the contextual and cultural nuances of sign language (Line 520).**
> > >
> > > **A:** Sign language translation faces several challenges, including culturally specific gestures, the importance of facial expressions and body language, regional dialects, and context dependence. Gestures may vary across cultures, and facial expressions and body language further complicate translation. More challenging is that many sign language words with vastly different meanings have strikingly similar gestures, which poses a significant challenge to the model's ability to distinguish them. Additionally, the meaning of the same gesture can change depending on context, requiring models to understand multiple sign languages and their cultural nuances.
> > >
> > > **Q12.	The example of RAG.**
> > >
> > > **A:** Queries typically relate to themes or keywords from the initial translations [ref7,8]. For example, the initial output for PHOENIX-2014T is "am tag vor allem im norden regen." Using RAG, we verify and refine it with the query: "Check the accuracy and grammar of 'am tag vor allem im norden regen' in German."
> > >
> > > [ref7] Retrieval-augmented generation for knowledge-intensive NLP tasks. NeurIPS 2020.
> > > [ref8] When not to trust language modeals: Investigating effectiveness of parametric and non-parametric memories. ACL 2023.
> > >
> > > **Q13.	The meanings of ''w/o RAG'', ''pre-SignLLM'', and ''post-SignLLM''.**
> > >
> > > **A:** ''w/o RAG'' indicates that the RAG module is not used in our pipeline. ''Pre-SignLLM'' and ''post-SignLLM'' refer to the execution stages of RAG in our pipeline. RAG is executed before SignLLM in 'Pre-SignLLM' and after in 'Post-SignLLM' (**Lines 502-503**).
> > >
> > > **Q14. The source of the derived training data （Line 407）using keypoints and image frames in the diagram.**
> > >
> > > **A:** Several papers in your references express this in the same way.

---

### Official Review · Reviewer_pCAa · 2024-11-03

**Soundness:** 3
**Presentation:** 4
**Contribution:** 4
**Rating:** 8
**Confidence:** 4

**Summary:**

This is a well written contribution to the field of automatic sign-language translation. The work build on state of the art LLM modelling and unsupervised deep learning representations. They adopt latest hierarchical VQ-VAE techniques in multi-modal generative AI from video and apply it to learn discrete representation of sign codes. They adopt finetune techniques of solid baseline LLMs like T5 and finally they demonstrate the benefit of applying RAG techniques to gorund the model and improve the baseline LLM response.

**Strengths:**

This paper is a well rounded contribution, where the motivation, state of the art techniques and limitations are well presented. the work introduces the approach in a very clear in detail manner, specially the baseline techniques like VQ-VAE and the steps they build the system and approach. Figures are self explainable and authors make a good justification and introduction of RAG and how it is apply in their use case.

I value the rigurosity of authors presenting the evaluation with statistical significant test and solid demonstration of improved results over baseline and the attempt to share the parameters and configurations in the appendix to make the approach reproducible.

This is a strong paper that moves the needle in the technology apply to sign language and can potentially democratize the incorporation of it at scale into multiple products and services that could have a big impact in part of the population.

**Weaknesses:**

This is a good paper, but would have been great if the authors deep deeper into the potential and emerging properties of training the system based on GPT LLMS. Sign languages have also regionaliations, dialects and variations. The study lacks that analysis and how transfer learning can be applied in order to build a Sign-Language Foundational model.

Last but not least, as a complement beyond objective evaluations, would have been good in authors present a solid evaluation with human in the loop (Sign-Language users)

**Questions:**

a) Why authors didn't break down the analysis and study across sign and text based languages?
b) Why authors didn't conduct and human evaluation study?
c) What is the impact of speakers diversity, camera angles, etc in the accuracy of the codebook and final results?

---

> ### Author Response · Authors · 2024-11-26
>
> We sincerely thank the reviewer for the time and constructive feedback. We address the main concerns below.
>
> **Q1.  Further discussion on the potential and emerging properties of the GPT-based LLMS training system.**
>
> **A:** Thank you for raising an excellent point. Regarding the significant potential of GPT-based LLMS in sign language translation, we consider that it primarily manifests in: (1) the ability of LLMS to effectively recognize and generate complex gesture sequences, such as those in intricate conversational scenarios; (2) an enhanced understanding of the context and cultural meanings behind gestures; and (3) the integration of visual and linguistic information, particularly auditory data, through multimodal learning. This integration facilitates precise semantic conversions and natural interactions between spoken and sign language users. Additionally, these models support personalized training to adapt to various sign language dialects and individual expression styles, offering customized services. These are also the research goals we aim to achieve in our future work.
>
> **Q2.  Discussion on regionalization, dialects, and variants of sign language, and how to apply transfer learning to develop foundational models.**
>
> **A:** Thank you for the opportunity to expand on our task.  In sign language translation, developing a foundational model with transfer learning may explore several directions: Firstly, training the base model on data-rich sign language datasets and fine-tuning it for specific dialects to reduce reliance on extensive labeled data. Secondly, leveraging world knowledge embedded in resource-rich languages can enable the model to understand sign language expressions across different cultures and regions, particularly aiding languages with limited initial corpora.
>
> **Q3.  Suggesting the addition of human-centric reliability assessments (by sign language users) to complement objective evaluations.**
>
> **A:** Thank you for your constructive suggestion. Integrating a human-centered reliability assessment, including feedback from sign language users, would improve our evaluation of the translation model's reliability. However, current practical constraints (such as difficulties in finding sufficient numbers of professional sign language users) pose significant challenges for us. Additionally, we employ the widely adopted and validated evaluation metrics BLEU and ROUGE, and will provide the visualization code for our sign action reconstruction. We will explore new evaluation methods and keep your suggestions in mind for future improvements and optimizations. Thank you once again for your valuable comments.
>
>
> **Q4.  Reasons for the absence of analysis and research on sign and text-based languages.**
>
> **A:**We appreciate your careful review. The analysis and study of sign and text-based languages, such as understanding structural differences between sign and spoken languages, exploring translation issues, and representing sign language in text form, have been thoroughly examined in existing literature [ref1,2,3,4]. In addition, relevant analyses of the How2Sign and PHOENIX-2014T datasets have been thoroughly studied in some literature [ref5,6,7]. This paper, however, focuses on improving the performance of sign language translation, with an emphasis on performance-related metrics.
>
> [ref1] Sign language structure: An outline of the visual communication systems of the american deaf. Journal of deaf studies and deaf education, 2005.
>
> [ref2] Language deprivation and deaf mental health. Routledge, 2018.
>
> [ref3] American sign language: The phonological base. Sign language studies, 1989.
>
> [ref4] Toward a phonetic representation of signs: Sequentiality and contrast. Sign Language Studies, 2011.
>
> [ref5] Neural Sign Language Translation. CVPR 2018.
>
> [ref6] How2Sign: A Large-Scale Multimodal Dataset for Continuous American Sign Language. CVPR 2021.
>
> [ref7] A survey on Sign Language machine translation. 2023.
>
> **Q5.  The impact of speaker diversity and camera angles on codebook accuracy and final outcomes.**
>
> **A:** Thank you for your constructive feedback. As the How2Sign and PHOENIX-2014T datasets include multiple signers, the impact of speaker diversity on translation performance is already incorporated into our experiments. Moreover, by using human keypoints as training data, we naturally remove much identity-related information, which helps mitigate the impact of different signers on performance. Regarding the impact of camera angles, we have explored this aspect using the How2Sign dataset, which includes side-view data, and found no significant improvement in performance. This is mainly because, in the side view, crucial key points like the elbow and hand often overlap, appearing as a single point. For example, keypoints from the side view of the palm can only be observed at the pinky, while the other fingers are obscured.

---

### Official Review · Reviewer_5XfD · 2024-11-04

**Soundness:** 2
**Presentation:** 2
**Contribution:** 2
**Rating:** 3
**Confidence:** 4

**Summary:**

The authors propose an approach to sign language translation (from sign language video to text in a spoken language) that combines several elements:  (1) a sign tokenizer based on VQ-VAE to learn a vocabulary of sign segments corresponding to short multi-frame sequences, (2) a language model based on FLAN-T5 fine-tuned to accept a mixed vocabulary of sign and text tokens, and (3) retrieval augmentation to improve the quality of the results.  The paper claims improved performance on two commonly used benchmarks, How2Sign (for American Sign Language to English) and PHOENIX-2014T (for German Sign Language to German).  The paper also provides several ablation studies showing the effect of codebook and model sizes, sign tokenizer variations, and retrieval augmentation.

**Strengths:**

+ The idea of learning a tokenization for sign language video and combining it with text tokens is compelling, and the use of a VQ-VAE-based approach for this seems sensible.

+ This work represents the first use of RAG for sign language translation, as far as I know.

**Weaknesses:**

- The results do not seem to outperform SOTA results as claimed.  Specifically, Rust et al. 2024 (cited in the introduction but not included in the experiments) obtains BLEU-{1,2,3,4} scores of {38.9, 24.1, 16.4, 11.6} on How2Sign in what looks like the same setting, compared to the best results in this paper of {36.38, 21.80, 13.76, 9.88}.  (Please correct me if I'm wrong, of course.)

- The description of the approach is hard to follow, with many missing details.  See questions below.

- The paper contains some questionable statements about sign language and sign language research.  See details below.

- It lacks analysis of the learned tokens (e.g. do they tend to correspond with signs?).  This isn't strictly necessary, and is challenging for a dataset like How2Sign without glosses, but it should be doable for PHOENIX-14T and would be very helpful.

- In general, the writing needs improvement.  It is frequently repetitive/unclear/ungrammatical.

**Questions:**

Model details:

- I understand that the input m^{1:M} is a sequence of estimated keypoints.  What method is used for keypoint estimation, and what data was it trained on?

- Do I understand correctly that the representation (z_u, z_h) that is quantized to produce the sign tokens is simply the keypoints passed through a 1D convolution layer?  Have you tried using a more complex encoder to produce z_u and z_h?

- Have you compared FLAN-T5 to, say, a plain text translation model, e.g. T5 as used in Rust et al. 2024 cited in the paper, which would not require a prompt or combined sign+text vocabulary?

- In Section 3.3, what is the query for retrieval augmentation, and what are the "chunks"?

- What do pre-SignLLM and post-SignLLM mean?


Other details/questions:

- What does "VRG" stand for?

- The paper often describes the task as translation from sign language to spoken language.  I think it's worth stating more clearly that the output is written, although I understand that the intention is that it is the written form of a spoken language.

- What is meant by "phonetic disparity between sign and spoken languages"?

- What are "modish benchmarks"?  (Typo?)

- "Sign language ... possesses its own grammar, word formation, and lexicon."  What is meant by "word formation"?  Does this refer to morphology of sign languages, the process of word formation, or something else?

- "These differences, especially in word order, make transcription between the two languages complex."  I assume that "transcription" should be "translation."  Also, differences in word order are common between spoken/written languages as well.  Why do word order differences make translation from a sign language any more complex than translation between spoken/written languages?

- I assume the authors are aware of this, but the above two sentences suggest that there is a single sign language, whereas in fact there are of course many, so it is worth re-wording to reflect that.

- This statement seems incorrect to me:  "Previous research generally divides SLT into two distinct tasks:  Sign2Notation ... and Notation2Text."  This doesn't seem to be true about most recent sign language translation approaches, both for the datasets/language pairs used here and for others.  Some examples from the recent literature (some cited in the paper, but most not):

D. Li et al., "TSPNet: Hierarchical feature learning via temporal semantic pyramid for sign language translation," NeurIPS 2020.

B. Shi et al., "Open-Domain Sign Language Translation Learned from Online Video," EMNLP 2022.

L. Tarrés, "Sign language translation from instructional videos," CVPR 2023.

K. Lin et al., "Gloss-Free End-to-End Sign Language Translation," ACL 2023.

P. Rust et al. "Towards Privacy-Aware Sign Language Translation at Scale," arXiv:2402.09611, 2024.

---

> ### Author Response · Authors · 2024-11-26
>
> We thank the reviewer for the time and feedback. We address the main questions below.
>
> **Q1. The results do not seem to outperform SOTA results as claimed.**
>
>   **A:** We respectfully disagree. Our results surpass the SOTA under the same settings.
> The results you mentioned in [ref1] are obtained using the Youtube-ASL dataset [ref2] for training, not How2Sign. Under the same experimental setup, [ref1] reports BLEU-{1,2,3,4} scores of $30.2$, $16.7$, $10.5$, and $7.0$ (**Table 2 in [ref1]**). Hence, our approach demonstrates superior performance compared to [ref1].
> Additionally, we follow a standard task setup for sign language translation, focusing on direct sign-to-spoken language translation. By contrast, the main goal of [ref1] is to mitigate privacy risks in large-scale web-crawled datasets, as reflected in the title of [ref1]. Thus, a direct comparison between our method and [ref1] is not appropriate.
>
>  [ref1] Towards Privacy-Aware Sign Language Translation at Scale. ACL 2024.
>  [ref2] YouTube-ASL: A Large-Scale, Open-Domain American Sign Language-English Parallel Corpus. NeurIPS 2023.
>
> **Q2.	What method is used for keypoint estimation, and on what dataset is it trained?**
>
> **A:** The keypoint estimation method used is Openpose, a widely employed tool for human pose estimation. The keypoint data we use is provided by the publicly available How2Sign raw dataset [ref3], not extracted by us.
>
> [ref3] How2sign: A large-scale multimodal dataset for continuous american sign language. CVPR 2021.
>
> **Q3.	Lack of analysis on learned tokens (*e.g.* do they tend to correspond with signs?).**
>
> **A:** Due to the ambiguity in your statement, we are unable to fully understand which aspect of 'the analysis on learned tokens' you are referring to. Therefore, we are assuming that you are referring to an analysis of whether VQ-VAE can accurately reconstruct the output sign sequence. The ability of VQ-VAE to accurately replicate the input after training has been frequently validated across nearly all deep learning-related fields. Additionally, we reference [ref4] to implement sign reconstruction visualization, which provides a clear comparison of the reconstruction.
>
> [ref4] MotionGPT: Human Motion as a Foreign Language. NeurIPS 2023.
>
> **Q4. The network structure for generating sign tokens and whether more complex encoders have been tried.**
>
> **A:** You are right, our sign encoders are built with 1D convolutions, and we have tried using more complex encoders, such as transformers. During our exploration, more complex encoders did not yield significant performance gains but significantly increased training time and debugging difficulty. Therefore, we ultimately chose to use 1D convolutions for the sign encoders.
>
> **Q5.	Have you compared FLAN-T5 to a basic text translation model like T5, as used in [ref1], which avoids prompts and combined sign+text vocabularies?**
>
> **A:** I would like to clarify that [ref1] does not explicitly state that the T5 model [ref5] used avoids prompts. On the contrary, using prompts is the default setting when training T5. Additionally, prompts have been shown to be an effective training strategy in nearly all large-scale model studies, with minimal adverse effects.
>
> The main reasons for not comparing FLAN-T5 [ref6] with T5 are as follows: Firstly, FLAN-T5 and T5 are language models based on the same architecture; FLAN-T5 does not alter the original T5 design. Secondly, FLAN-T5 integrates instruction tuning with extensive fine-tuning data, boosting its robustness and accuracy in multi-task and multilingual settings. We also briefly discussed the reasons for using FLAN-T5 in the manuscript (**Lines 191-193, 316-318**). For the above reasons, we conduct experimental analysis only on FLAN-T5.
>
> [ref5] Exploring the limits of transfer learning with a unified text-to-text transformer. JMLR, 2020.
> [ref6] Scaling Instruction-Finetuned Language Models. JMLR, 2024.
>
> **Q6.In Section 3.3, what is the query for retrieval augmentation, and what are the "chunks"?**
>
> **A:** Queries typically relate to themes or keywords from the initial translations [ref7,8]. For example, the initial output for PHOENIX-2014T is "am tag vor allem im norden regen." Using RAG, we verify and refine it with the query: "Check the accuracy and grammar of 'am tag vor allem im norden regen' in German." 'Chunks' are retrieved information segments used to support generation tasks [ref7,8].
>
> [ref7] Retrieval-augmented generation for knowledge-intensive NLP tasks. NeurIPS 2020.
> [ref8] When not to trust language modeals: Investigating effectiveness of parametric and non-parametric memories. ACL 2023.
>
> **Q7.The meanings of ''w/o RAG'', ''pre-SignLLM'', and ''post-SignLLM''.**
>
> **A:** ''w/o RAG'' indicates that the RAG module is not used in our pipeline. ''Pre-SignLLM'' and ''post-SignLLM'' refer to the execution stages of RAG in our pipeline. RAG is executed before SignLLM in 'Pre-SignLLM' and after in 'Post-SignLLM' (**Lines 502-503**).

---

> > ### Author Response · Authors · 2024-11-26
> >
> > **Q8.	What does "VRG" stand for?**
> >
> > **A:** VRG is an alias used in our paper for convenience, representing sign language translation LLMs enhanced with VQ-VAE and RAG.
> >
> > **Q9.	Discussion on whether the task should be described as translating sign language to written text instead of spoken language.**
> >
> > **A:** Most existing papers on sign language translation, particularly all the references you mentioned [ref1, 9, 10, 11, 12], describe it as a translation between sign language and spoken language. Therefore, we consider it is unnecessary to replace spoken language with written text.
> >
> > [ref9] Open-Domain Sign Language Translation Learned from Online Video. EMNLP 2022.
> >
> > [ref10] Sign language translation from instructional videos. CVPR 2023.
> >
> > [ref11] Gloss-Free End-to-End Sign Language Translation. ACL 2023.
> >
> > [ref12] TSPNet: Hierarchical feature learning via temporal semantic pyramid for sign language translation. NeurIPS 2020.
> >
> > **Q10.	The meaning of ''phonetic disparity between sign and spoken languages''.**
> >
> > **A:** The phrase "phonetic disparity between sign and spoken languages" refers to the differences in how meaning is conveyed, a well-established concept in sign language research [ref13,14,15]. Spoken languages rely on vocal sounds to convey meaning, while sign languages use visual-spatial elements such as gestures, facial expressions, and body movements. As a result, the phoneme inventories of spoken and sign languages do not fully overlap, and certain phonemes may not have direct equivalents in both types of languages. This disparity arises because sign language does not rely on sound or hearing; instead, it uses visual and spatial modes of communication. Consequently, sign languages have distinct linguistic structures and phonological systems that are fundamentally different from those of spoken languages.
> >
> > [ref13] American sign language: The phonological base. Sign language studies, 1989.
> >
> > [ref14] Toward a phonetic representation of signs: Sequentiality and contrast. Sign Language Studies, 2011.
> >
> > [ref15] The phonological organization of sign languages. Lang, Linguistics Compass, 2012.
> >
> > **Q11.	What are "modish benchmarks"? (Typo?)**
> >
> > **A:** This is not a typo. ''Modish benchmarks'' refers to the latest and most popular evaluation datasets; in this paper, it specifically denotes the How2Sign and PHOENIX-2014T datasets.

---

> > > ### Author Response · Authors · 2024-11-26
> > >
> > > **Q12.	Does ''word formation'' refer to the morphology of sign language, the process of word formation, or something else?**
> > >
> > > **A:** ''Word formation'' in sign language involves constructing words using a visual-spatial approach, unlike spoken languages. Words in sign language are created through gestures, facial expressions, and body movements, with key elements like handshapes, placement, movement, and palm orientation shaping their expressive vocabulary [ref13,14,15,16].
> > >
> > > [ref16] Interaction of morphology and syntax in American Sign Language. Routledge, 2016.
> > >
> > > **Q13. Discussion of the sentence “These differences, especially in word order, make transcription between the two languages complex.”**
> > >
> > > **A:** First, I would like to clarify that this statement means: ''The differences between languages, especially in word order, make transcription from one language to another complex. In other words, due to the differences in grammatical structure, particularly word order, converting content from one language to another involves not only matching vocabulary but also considering the differences in sentence structure and word order.'' This sentence does not claim that only word order differences make translation from a sign language more complex.
> > > In that way, the grammatical differences between sign and spoken languages make translation complex because sign language conveys emotional and contextual information through gestures, facial expressions, and body language, while spoken language typically relies on detailed verbal descriptions. Thus, translating sign language requires not only grammatical adjustments but also capturing its subtle nuances [ref13,14,15,16].
> > >
> > > **Q14.	Suggestions to rephrase and acknowledge the variety of sign languages around the world.**
> > >
> > > **A:** We respectfully disagree. There are approximately 200 countries globally, each with its own sign language. Consistent with the standard task settings in [ref1-3, 7-12], we conduct our sign language translation within the confines of one country’s language, a standard setting in this field.
> > >
> > > **Q15. Discussion on dividing sign language translation into Sign2Notation and Notation2Text processes.**
> > >
> > > **A:**   We disagree with respect. Given the formidable challenge of sign language translation, mainstream papers  [ref17,18,19,20] do not translate directly from sign actions to spoken language but require an intermediary step to achieve higher translation accuracy. For example, it may involve glosses or HamNoSys ( a lexical Sign language notation), proceeding from sign to gloss to text, or from gloss to HamNoSys to text. Therefore, it is appropriate to categorize sign language translation into the steps of Sign2Notation and Notation2Text (**Lines 53-72**). Here, 'Notation' refers to any intermediary marker, such as gloss, HamNoSys, or other identifiers.
> > >
> > >  [ref17] Neural Sign Language Translation. CVPR 2018.
> > >
> > >  [ref18] Sign language transformers: Joint end-to-end sign language recognition and translation. CVPR 2020.
> > >
> > >  [ref19] Spatial-Temporal Multi-Cue Network for Sign Language Recognition and Translation. TMM, 2022.
> > >
> > >  [ref20] Ham2Pose: Animating Sign Language Notation into Pose Sequences. CVPR 2023.

---

> > ### Comment · Reviewer_5XfD · 2024-11-30
> >
> > Thank you for your responses.  Some of them help to clarify things for me.  I'll respond to the more important points:
> >
> > Q1. The results do not seem to outperform SOTA results as claimed.
> > A: We respectfully disagree...
> >
> > I understand your motivation for comparing to prior work that used the same training setting.  However, the meaning of "same setting" is a bit murky these days, since most systems (including yours) use components that have been pre-trained on various datasets that differ from those of prior work.  Arguably, the only meaningful definition of "SOTA on test set X" is "the best existing result on test set X", regardless of what the system was trained on (but without training on test set X).  Alternatively, one could talk about "SOTA on test set X when trained on set(s) Y", or something like that.  In any case, the paper should make clear that better results exist, while clarifying how they were obtained.  Overall, I think a more accurate description of your How2Sign results is that they are promising, and might beat the best existing results if trained on more data.
> >
> > Q3. Lack of analysis on learned tokens (e.g. do they tend to correspond with signs?).
> > A: Due to the ambiguity in your statement, ...
> >
> > I was asking for any analysis of how meaningful the learned tokens are.  Measuring the reconstruction performance of the VQ-VAE wouldn't really get at this.  As I mentioned, an interesting question is whether the tokens correspond to signs.  This could be analyzed qualitatively, for example, by presenting example images corresponding to a given token.
> >
> > Q6, Q7
> >
> > I'm afraid I do not follow your answers, although I am familiar with RAG.  The example you provide ("Check the accuracy...") seems like a query one can give to any LLM for post-editing, not necessarily a RAG-based model.  Some examples in the appendix might help.  For example, for each of "w/o RAG", "pre-SignLLM", and "post-SignLLM", you could provide the initial output, the query to the RAG system, the retrieved chunks, and the final output (for the cases where RAG is used).

---

> ### Comment · Reviewer_5XfD · 2024-11-30
>
> Thank you for the responses.  Regarding the various points about word order, word formation, phonetics and phonology:
>  Unfortunately the responses do not clarify things in my opinion.  I still find these descriptions about the unique aspects of sign language to be misleading.  While these are relatively minor points, all together I am afraid they may add to the many misconceptions about sign language that exist in the literature.  In my view, if all of these claims were simply removed from the paper, it would improve the paper's quality.

---

### Author Response · Authors · 2024-11-26

To all reviewers:

We extend our sincere gratitude to all the reviewers for their valuable efforts. Based on your feedback, we have made revisions to our paper. Below, we summarize the main changes in the paper.  Before we present the revised parts, we would like to highlight the importance and novelty of our work, which has been recognized by several reviewers. We introduce VRG-SLT, a novel sign language translation model that integrates hierarchical VQ-VAE and LLMs, enhanced by RAG for improved translation. The key contributions are: (1) A collaborative hybrid model where sign movements are treated as a unique language and jointly trained with text using LLMs; (2) A sign-tokenizer that captures both upper body and hand trajectory characteristics, leveraging a hierarchical structure to handle complex and diverse movements; (3) The integration of RAG, enabling the retrieval and incorporation of relevant knowledge for more accurate and content-rich outputs. Finally, our model achieves superior performance on standard benchmarks. We believe our contributions hold significant potential for the community. We address all reviewers' concerns and comments in a point-by-point response below.

**The major changes are as follows:**
1. In response to Reviewer pCAa, we discuss issues such as the GPT-based LLM training system, the use of transfer learning for developing foundational models in different contexts, and the impact of multiple signers and different camera angles on performance.
2. We further discuss with the reviewer RLDc how the RAG model addresses sparsity in sign language translation and how it handles subtle changes in meaning during text generation.
3. We reiterate the novelty and contributions of our method and analyze the differences from the references mentioned by Reviewer 5XfD and x9jx.
4. We provide explanations for certain words, phrases, and technical terms, along with definitions from the Oxford English Dictionary to aid understanding, based on the majority of comments from Reviewer 5XfD and x9jx.
5.  We provide explanations for certain words and phrases, along with definitions from the Oxford English Dictionary to aid understanding, based on the comments of Reviewer 5XfD and x9jx.

Sincerely,

Authors.

---

### Note · Authors · 2025-01-23

I have read and agree with the venue's withdrawal policy on behalf of myself and my co-authors.